# Low-Rank Learning by Design: the Role of Network Architecture and Activation Linearity in Gradient Rank Collapse

## Abstract

Our understanding of learning dynamics of deep neural networks (DNNs) remains incomplete. Recent research has begun to uncover the mathematical principles underlying these networks, including the phenomenon of "Neural Collapse", where linear classifiers within DNNs converge to specific geometrical structures during late-stage training. However, the role of geometric constraints in learning extends beyond this terminal phase. For instance, gradients in fully-connected layers naturally develop a low-rank structure due to the accumulation of rank-one outer products over a training batch. Despite the attention given to methods that exploit this structure for memory saving or regularization, the emergence of low-rank learning as an inherent aspect of certain DNN architectures has been under-explored. In this paper, we conduct a comprehensive study of gradient rank in DNNs, examining how architectural choices and structure of the data affect gradient rank bounds. Our theoretical analysis provides these bounds for training fully-connected, recurrent, and convolutional neural networks. We also demonstrate, both theoretically and empirically, how design choices like activation function linearity, bottleneck layer introduction, convolutional stride, and sequence truncation influence these bounds. Our findings not only contribute to the understanding of learning dynamics in DNNs, but also provide practical guidance for deep learning engineers to make informed design decisions.

## 1 Introduction

Deep Neural Networks (DNNs) continue to achieve state-of-the-art performance on a number of complex data sets for a diverse array of tasks; however, modern DNN architectures are notoriously complex, with millions of parameters, nonlinear interactions, and dozens of architectural choices and hyper-parameters which can all significantly affect model performance. Internal complexity and a lack of thorough theoretical groundwork has given DNNs a reputation as "black box" models, where architectures may excel or fail on a given problem with relatively little indication how their structure facilitated that performance. Engineering a neural network that works well on a particular problem can often take the form of fairly arbitrary and exhaustive model tweaking, and even in cases where researchers systematically perturb particular settings, the primary explanations of performance come down to observing minor changes in performance evaluation metrics such as loss, accuracy/precision, dice-scores or other related metrics. In this work, we examine a particular emergent phenomenon in deep neural networks—the collapse of gradient rank during training; however, we take a theory-first approach, examining how bounds on gradient rank collapse appear naturally and deterministically as a result of particular architectural choices such as bottleneck layers, level of nonlinearity in hidden activations, and parameter tying.

This work is part of a growing body of theoretical research studying the dynamics of learning in deep neural networks. Beginning from first principles, Saxe et al. [2013] provided a foundational work on exact solutions to nonlinear dynamics which emerge in fully-connected networks with linear activations, which has inspired a body of related work on simple networks with various architectural or learning setups such as parent-teacher interactions [Goldt et al., 2019a], online learning and overparametrization [Goldt et al., 2019b], and gated networks [Saxe et al., 2022]. This work on theory and dynamics has also been extended to studying high-dimensional dynamics of generalization error [Advani et al., 2020], emergence of semantic

representations [Saxe et al., 2019], and other phenomemon which can be first characterized mathematically and observed empirically. Our work in this paper follows this tradition in the literature of beginning with mathematical principles which affect learning dynamics in simple networks, and demonstrating how these principles emerge in practical scenarios.

An additional related body of work studies the phenomemon of Neural Collapse [Papyan et al., 2020; Kothapalli, 2023], in which deep classifier neural networks converge to a set of rigid geometrical constraints during the terminal phase of training, with the geometry of the output space determined exactly by the number of classes (i.e., the rank of the output space). This neural collapse phenomenon has been thoroughly studied as emerging in constrained [Yaras et al., 2022] and unconstrained feature settings [Mixon et al., 2020; Zhu et al., 2021; Tirer & Bruna, 2022], with various loss functions [Lu & Steinerberger, 2020; Han et al., 2021; Zhou et al., 2022b], under transfer learning regimes [Galanti et al., 2021], class imbalance scenarios [Thrampoulidis et al., 2022], and exemplifying affects on the loss-landscapes [Zhou et al., 2022a]. This growing body of work suggests that geometric constraints during learning influence a number of desirable and undesirable deep learning behaviors in practice.

Like the works on neural collapse, we are interested in geometric constraints to learning; however, we follow the example of Saxe et al. and begin our theoretical examination with simple linear networks, showing how we can expand on simple constraints of batch size (previously discussed by Papyan [2018]) to constraints dependent on a number architectural features such as bottlenecked layers, parameter-tying, and level of linearity in the activation. Our work invites the study of network-wide geometric constraints, and while we do not dive into training dynamics in this work, we are able to set a stage which bounds dynamics, hopefully clearing the way for further dynamical analysis in the style of Saxe, Tishby and others. In short, our work provides a study of the architectural constraints on the dimension of the subpaces in which dynamics such as Neural Collapse or those studied by Saxe et al. can occur.

Finally, our work studying the affect of a particular nonlinear activation and its level of linearity stands out from the existing work on purely linear networks and neural collapse in linear classifiers. Although nonlinear activations introduce some complexity to analysis, we can draw from some previous work on analyzing the spectrum of ReLU activations [Dittmer et al., 2019], extending the analysis in that work to its natural consequences with Leaky-ReLU activations and even deriving explicit bounds for numerical estimation of rank which require only singular values of the underlying linear transformations.

Our derivation of upper bounds on gradient dynamics during training also has implications for distributed models which utilized low-rank decompositions for efficient communication. For example, PowerSGD [Vogels et al., 2019] and distributed Auto-Differentiation [Baker et al., 2021] compute a low-rank decomposition of the gradient prior to transfer between distributed data-collection sites. Our theoretical results here can help to provide insights into how high of a rank may be needed to preserve model performance between the pooled and distributed cases, or how much information may be lost when a lower-rank decomposition is used for communication efficiency.

Our primary results in this work are theoretical; however, we perform a number of empirical verifications and demonstrations which verify our theoretical results and study the implications of our derived bounds on gradient rank for various archictural design choices. In sum, the contributions here include:

(a) an upper bound on the rank of the gradient in linear networks;
(b) upper bounds on the rank of the gradient in linear networks with shared parameters, such as RNNs and CNNs;
(c) extension of previous work on ReLU Singular Values to study the effect of Leaky-ReLUs and the level of linearity on the upper bound of rank;
(d) empirical results on numerical data which verify our bounds and implicate particular architectural choices;
(e) empirical demonstration of theoretical implications on large-scale networks for Computer Vision and Natural Language Processing.

In addition, there appear to be natural extensions (left for future work) to rank dynamics during training, explicit connections to neural collapse, and implications for rank effects of other architectural phenomena such as dropout layers, batch norms, and more.

## 2 Theoretical Methods

It is our primary theoretical goal to clearly demonstrate how common architectural choices enforce bounds on the rank of training statistics in deep neural networks. For the time being, we confine our discussion to first-order optimization methods, namely gradient descent and its various flavors. We utilize Reverse-Mode Auto-Differentiation (AD) as our primary theoretical framing device, with low-rank gradients initially appearing as a consequence of batch gradient descent. In §2.1, we introduce our notation and formalize the groundwork which informs our theoretical results in §3.

After the formal introduction to reverse-mode AD, our theoretical results are generated first by following the precedent in [Saxe et al., 2013] of beginning with models with purely linear activations. Utilizing linear activations provides us with the ability to directly apply the machinery of linear algebra, and we derive upper bounds on gradient rank which utilizes only architectural choices in model design such as bottleneck layers and parameter tying.

Following our derivation of gradient bounds for fully-linear models, we extend our theoretical analysis to cover ReLU and Leaky-ReLU nonlinear activations. We briefly connect our results on linear-activated models to previous theoretical work which defined so-called "ReLU Singular Values" [Dittmer et al., 2019]. We then shift our focus to studying how the application of Leaky-ReLU nonlinearities with different negative slopes can affect the numerical approximation of singular values in deep neural networks in practice.

### 2.1 Reverse-Mode Auto-Differentiation

We will define a simple neural network with depth $L$ as the operator $\Phi(\{\mathbf{W}_i\}_{i=0}^L, \{\mathbf{b}_i\}_{i=0}^L, \{\phi_i\}_{i=0}^L) : \mathbb{R}^m \to \mathbb{R}^n$. This is given a set of weights $\{\mathbf{W}_1, \ldots, \mathbf{W}_L\}$, bias variables $\{\mathbf{b}_1, \ldots, \mathbf{b}_L\}$, and activation functions $\{\phi_1, \ldots, \phi_L\}$, where each function $\phi_i : \mathbb{R}^n \to \mathbb{R}^n$ is an element-wise operator on a vector space.

Let $\mathbf{x} \in \mathbb{R}^m$ be the input to the network and $\mathbf{y} \in \mathbb{R}^n$ be a set of target variables. We then define $z_i$ and $a_i$ as the *internal* and *external* activations at layer $i$, respectively, given by a recursive relation:

$$\mathbf{z}_0 = \mathbf{x} \qquad\qquad \mathbf{z}_i = \mathbf{W}_i \mathbf{a}_{i-1} + \mathbf{b}_i$$
$$\mathbf{a}_0 = \mathbf{z}_0 \qquad\qquad \mathbf{a}_i = \phi_i(\mathbf{z}_i)$$

To define the sizes of the hidden layers, we have $\mathbf{W}_i \in \mathbb{R}^{h_{i-1} \times h_i}$, with $h_0 = m$ and $h_L = n$. We note then that $\mathbf{z}_i, \mathbf{a}_i$ are column vectors in $\mathbb{R}^{h_i}$.

Let $\mathcal{L}(\mathbf{y}, \mathbf{a}_L) : \mathbb{R}^n \to \mathbb{R}$ be a loss function which measures the *error* of the estimate of $\mathbf{y}$ at $\mathbf{a}_L$. The gradient update for this loss, computed for the set of weights $\mathbf{W}_i$ can be written as the outer-product

$$\nabla_{\mathbf{W}_i} = \mathbf{a}_{i-1} \delta_i^\top$$

where $\delta_i$ is the partial derivative of the output at layer $i$ w.r.t its input. At the output layer $L$, $\delta_L$ is computed as

$$\delta_L = \frac{\partial \mathcal{L}}{\partial \mathbf{a}_L} \odot \frac{\partial \phi_L}{\partial \mathbf{z}_L}$$

and subsequent $\delta_i$ are computed as

$$\delta_i = \delta_{i+1} \mathbf{W}_{i+1} \odot \frac{\partial \phi_i}{\partial \mathbf{z}_i}.$$

These definitions are all given for $x$ as a column vector in $\mathbb{R}^m$; however, for standard batch SGD we compute these quantities over a batch of $N$ samples. If we rewrite the variables $\mathbf{x}$, $\mathbf{y}$, $\mathbf{z}_i$, $\mathbf{a}_i$ and $\delta_i$ as matrices

$\mathbf{X} \in \mathbb{R}^{N \times m}$, $\mathbf{Y} \in \mathbb{R}^{N \times n}$, $\mathbf{Z}_i, \mathbf{A}_i, \boldsymbol{\Delta}_i \in \mathbb{R}^{N \times h_i}$. The gradient can then be computed as the matrix-product

$$\sum_k^N \mathbf{a}_{n,i-1} \delta_{n,i}^\top = \mathbf{A}_{i-1}^\top \boldsymbol{\Delta}_i.$$

## 3 Theoretical Results

### 3.1 Bounds on Gradients in Linear Networks

First consider the set of neural networks where $\phi_i(\mathbf{x}) = \mathbf{x}$ is the identity operator $\forall i$. These networks are called "linear networks" or equivalently "multi-layer perceptrons" (MLP), and $\mathbf{Z}_i = \mathbf{A}_i, \forall i$. For these networks, the rank of the gradients has an exact bound. Trivially, for a given gradient $\nabla_{\mathbf{W}_i}$, the rank is

$$\mathrm{rank}(\nabla_{\mathbf{W}_i}) = \mathrm{rank}(\mathbf{Z}_{i-1}^\top \boldsymbol{\Delta}_i) \leq \min\{\mathrm{rank}(\mathbf{Z_{i-1}}), \mathrm{rank}(\boldsymbol{\Delta}_i)\}.$$

Since in linear networks we can easily compute $\mathbf{Z}_i = \mathbf{X} \prod_{j=1}^i \mathbf{W}_j$, we use a similar rule as for the gradient to compute the bound

$$\mathrm{rank}(\mathbf{Z}_i) \leq \min\{\mathrm{rank}(\mathbf{X}), \mathrm{rank}(\mathbf{W}_1), \ldots, \mathrm{rank}(\mathbf{W}_i)\}.$$

For the adjoint variable $\Delta_i$ we can use the fact that $\frac{\partial \phi}{\partial \mathbf{Z}_i} = \mathbf{1}$ where $\mathbf{1}$ is a matrix of ones in $\mathbb{R}^{N \times h_i}$ to derive a bound on the adjoint as

$$\mathrm{rank}(\boldsymbol{\Delta}_i) \leq \min\{\mathrm{rank}(\mathbf{W}_i), \mathrm{rank}(\mathbf{W}_{i+1}), \ldots, \mathrm{rank}(\mathbf{W})_L, \mathrm{rank}(\frac{\partial \mathcal{L}}{\partial \mathbf{Z}_L})\}.$$

Therefore, the bound for the gradient rank is

$$\begin{aligned}
\mathrm{rank}(\nabla_{\mathbf{W}_i}) &\leq \min\{\mathrm{rank}(\mathbf{Z_{i-1}}), \mathrm{rank}(\boldsymbol{\Delta}_i)\} \\
&\leq \min\{\mathrm{rank}(\mathbf{X}), \mathrm{rank}(\mathbf{W}_1), \ldots, \mathrm{rank}(\mathbf{W}_i), \mathrm{rank}(\mathbf{W}_{i+1}), \ldots, \mathrm{rank}(\mathbf{W})_L, \mathrm{rank}(\frac{\partial \mathcal{L}}{\partial \mathbf{Z}_L})\}
\end{aligned} \tag{1}$$

### 3.2 Bounds on Gradients in Linear Networks with Parameter Tying

We will begin our analysis with recurrent neural networks and Back-Propagation through Time (BPTT) [Rumelhart et al., 1986].

#### 3.2.1 Recurrent Layers

Let $\mathcal{X} \in \mathbb{R}^{N \times n \times T}$ be the $N$ samples of an $n$-dimensional variable over a sequence of length $T$ (over time, for example). We will set an initial hidden state for this layer as $\mathbf{H}_{i,0} \in \mathbb{R}^{N \times h_i}$.

Let $f_i : \mathbb{R}^{N \times h_{i-1} \times T} \to \mathbb{R}^{N \times h_i \times T}$ be the function given by a linear layer with a set of input weights $\mathbf{U} \in \mathbb{R}^{h_{i-1} \times h_i}$ and a set of hidden weights $\mathbf{V} \in \mathbb{R}^{h_i \times h_i}$. The output of this layer is computed as the tensor

$$f_i(\mathcal{X}) = \{\mathbf{H}_{t,i}\}_{t=1}^T = \{\phi_i(\mathbf{Z}_{t,i})\}_{t=1}^T = \{\phi_i(\mathbf{X}_t \mathbf{U}_i + \mathbf{H}_{t-1,i} \mathbf{V}_i)\}_{t=1}^T.$$

The adjoint variables computed during the backward-pass for $\mathbf{U}$ and $\mathbf{V}$) can be collected by simply computing the partial derivative as above, at each point in the sequence. Formally, we have

$$\mathcal{D}_i = \{\boldsymbol{\Delta}_{t,i}\}_{t=1}^T = \left\{ (\Delta_{t,i+1} \mathbf{U}_{i+1} + \boldsymbol{\Delta}_{t+1,i} \mathbf{V}_i) \odot \frac{\partial \phi_i}{\partial \mathbf{Z}_{t,i}} \right\}_{t=1}^T$$

where we have $\boldsymbol{\Delta}_{T+1,i} = \mathbf{0}$ for convenience of notation. In the case where the next layer in the network is not recurrent (for example if it is a linear layer receiving flattened output from the RNN), we can set $\boldsymbol{\Delta}_{t,i+1}$ to be the elements of $\boldsymbol{\Delta}_{i+1}$ which correspond to each timepoint $t$.

The gradients for $\mathbf{U}_i$ and $\mathbf{V}_i$ are then computed as the sum over the products of each element in the sequence

$$\nabla_{\mathbf{U}_i} = \sum_{t=1}^{T} \mathbf{X}_{t,i}^{\top} \mathbf{\Delta}_{t,i} \qquad\qquad \nabla_{\mathbf{V}_i} = \sum_{t=0}^{T-1} \mathbf{H}_{t,i}^{\top} \mathbf{\Delta}_{t,i}$$

where $\mathbf{X}_{t,i}$ is the input to hidden layer $i$ as time $t$, and $\mathbf{H}_{t,i}$ is the hidden state from layer $i$ at time $t$.

Bounds on the rank of these matrices amounts to an application of the identity:

$$\operatorname{rank}(A + B) \leq \operatorname{rank}(A) + \operatorname{rank}(B)$$

For the matrix $\mathbf{U}_i$ for example, the bound can just be written as a sum of the bound in equation 1:

$$\operatorname{rank}(\nabla_{\mathbf{W}_i}) \leq \sum_{t}^{T} \min\{\operatorname{rank}(\mathbf{X}_{t,i}), \operatorname{rank}(\mathbf{\Delta}_{t,i})\} \tag{2}$$

$$\leq \sum_{t}^{T} \min\{\operatorname{rank}(\mathbf{X}_{t,i}), \operatorname{rank}(\mathbf{U}_1), \ldots, \operatorname{rank}(\mathbf{U}_i), \operatorname{rank}(\mathbf{U}_{i+1}), \ldots, \operatorname{rank}(\mathbf{U})_L, \operatorname{rank}(\frac{\partial \mathcal{L}}{\partial \mathbf{Z}_{t,L}})\} \tag{3}$$

Fully linear RNNs are not typically implemented in practice; however, for the purpose of demonstrating how parameter-tying can improve with parameter-tying, the analysis may still prove helpful. The first thing to notice is that even for as small as $T = 2$, we reach the potential for full-rank gradients quite quickly (if the rank at each timepoint is half full-rank for eaxmple). Even in the degenerate case when the batch size is $N = 1$, $\nabla_{\mathbf{U}_i}$ and $\nabla_{\mathbf{V}_i}$ may reach rank $T$. Thus, the analysis of rank no longer depends much on the architecture beyond the number of timepoints chosen, and parameter-tying can affect rank-collapse that emerges from low-rank product bottlenecks in other layers. Rather, it will become of interest to look at correspondences between input such as temporal correlation in order to provide a clearer picture. We will leave this for future work.

### 3.2.2 Convolutional Layers

We can derive similar bounds to those for recurrent networks for the gradients over convolutional layers. At its core, the underlying principle of sharing gradients over a sequence remains the same; however, the particular architectural choices (stride, image size, kernel size, padding, dilation) which influence the length of this sequence are more numerous for convolutional layers. Thus, since the core idea is the same as for RNNs, our theoretical bounds on gradient rank in convolutional layers are included in our supplementary material.

### 3.3 Bounds on Gradients in Leaky-ReLU Networks

The bounds we provide on gradient rank in networks with purely linear activations builds off intuitive principles from linear algebra; however, the introduction of nonlinear activations confuses these notions. For a general nonlinear operator $\phi$, the notion of singular values which we obtain from linear algebra does not hold. Even though we can compute the SVD of the matrix $\phi_\alpha(\mathbf{Z})$ for a given layer with internal activations $\mathbf{Z}$, little can be initially said about the relationship of the this decomposition to the linear transformations which generated $\mathbf{Z}$. Thus, although we can still empirically compute the rank of the resulting matrix from the nonlinear transformation, it is not initially clear how rank-deficiency will propagate through a network as it will in the fully linear case.

Leaky-ReLU (Leaky Rectified Linear Unit) activations are defined as follows:

$$\phi(x) = \begin{cases} x & x > 0 \\ \alpha & \alpha x \leq 0 \end{cases}$$

where $\alpha \in \mathbb{R}$ controls the slope of the activation function in the negative domain. When $\alpha = 0$ Leaky-ReLUs are equivalent to ReLU activations. Importantly, we notice that Leaky-ReLUs, although globally nonlinear, are piece-wise linear functions at their core. This fact will allow us to show how Leaky-ReLU activations with different levels of nonlinearity affect numerical estimates of rank.

### 3.3.1 Numerical effect of Leaky-ReLUs on Rank

In this section, we analyze the numerical effect of Leaky-ReLU nonlinearities on the singular values of the internal activations $\mathbf{Z}_i = \mathbf{A}_{i-1}\mathbf{W}_i$. Our tactic will be to observe how the slope $\alpha$ can push the singular values of $\phi_\alpha(\mathbf{Z}_i)$ above a numerical threshold used for estimating rank.

For our numerical threshold, we use $\epsilon \cdot \max_i \sigma_i$, where $\epsilon$ is the machine-epsilon for floating-point calculations on a computer, defined such that $\epsilon \in \mathbb{R}, 0 < \epsilon << 1$. In practice, machine epsilon is determined by the computing architecture and numerical precision. For example, 32-bit floating-point numbers in PyTorch correspond with a machine epsilon of around $1 \times 10^{-7}$. $\{\sigma_i\}$ is the set of singular values of the matrix in question for which we aim to estimate rank.

To connect this numerical threshold to numerical rank, we can estimate the numerical rank of a matrix by determining whether or not each singular value falls above or below that threshold. Formally, we say that the $k$th singular value $\sigma_k$ does not contribute to our estimation of numerical rank if

$$\sigma_k < \epsilon \cdot \max_i \sigma_i. \tag{4}$$

If we can find such a $k$, we would therefore conclude that the rank of the matrix is $k - 1$. Otherwise, if all singular values fall above this threshold, the matrix is determined to be full rank.

For clarity, in the following derivation we will refer to an arbitrary layer in a feed forward network which has input $\mathbf{X} \in \mathbb{R}^{N \times m}$, weight matrix $\mathbf{W} \in \mathbb{>} \times \widetilde{\simeq}$, and internal activations $\mathbf{Z} \in \mathbb{R}^{N \times h}$ computed as $\mathbf{Z} = \mathbf{XW}$.

Let $\mathbf{D}_\alpha(\mathbf{Z}) \in \mathbb{R}^{N \times h}$ be the matrix with entries corresponding to the linear coefficients from the Leaky-ReLU activation applied to $\mathbf{Z}$. In other words, wherever an entry in $\mathbf{Z}$ is less than 0, the corresponding entry of $\mathbf{D}_\alpha$ will have an $\alpha$, and where $\mathbf{Z}$ is greater than 0, the corresponding entry of $\mathbf{D}_\alpha$ will be one. Using this construction, we notice that Leaky-ReLUiactivations can be written as the Hadamard (elementwise) product

$$\phi_\alpha(\mathbf{Z}) = \mathbf{D}_\alpha \odot \mathbf{Z} \tag{5}$$

Here we have exploited the piecewise-linearity of the Leaky-ReLU activation function in order to rewrite the nonlinear function as a producut which has useful identities as an linear algebraic operator.

From Zhan [1997], we have the inequality for the singular values of the Hadamard product:

$$\sum_{i=1}^{K} \sigma_i(\mathbf{D}_\alpha \odot \mathbf{Z}) < \sum_{i=1}^{K} \min\{c_i(\mathbf{D}_\alpha), r_i(\mathbf{D}_\alpha)\}\sigma_i(\mathbf{Z}_i) \tag{6}$$

where $c_1(\mathbf{D}_\alpha) \geq c_2(\mathbf{D}_\alpha) \geq \cdots \geq c_h(\mathbf{D}_\alpha)$ are the 2-norm of the columns sorted in decreasing order, and $r_i(\mathbf{D}_\alpha)$ are the 2-norm also in decreasing order. $\sigma_i(\mathbf{D}_\alpha \odot \mathbf{Z})$ is the $i$th singular value of the post-activation matrix $\mathbf{D}_\alpha \odot \mathbf{Z}$, and $\sigma_i(\mathbf{Z})$ is the $i$th singular value of the internal activations $\mathbf{Z}$. Here, we let $K$ denote the maximum number of singular values in $\mathbf{D}_\alpha \odot \mathbf{Z}$, and we note that for $\mathbf{Z}$ in particular $K = \min(N, m)$.

Because the inequality provided from Zhan et al. involves a sum, we cannot provide a one-to-one corresponence between singular values except for the largest, i.e. where $K = 1$ (since singular values are sorted by magnitude). Rewriting the inequality in equation 6, we get

$$\sigma_1(\mathbf{D}_\alpha \odot \mathbf{Z}) \leq \min\{c_1(\mathbf{D}_\alpha), r_1(\mathbf{D}_\alpha)\}\sigma_1(\mathbf{Z}_1). \tag{7}$$

Since our aim is to provide a bound like in equation 4, we combine equations 4 and 7 by saying the $i$th (post-activation) singular value will not contribute to the estimation of numerical rank if

$$\sigma_i(\mathbf{D}_\alpha \odot \mathbf{Z}) < \epsilon\sigma_1(\mathbf{D}_\alpha \odot \mathbf{Z}) \leq \epsilon \min\{c_1(\mathbf{D}_\alpha), r_1(\mathbf{D}_\alpha)\}\sigma_1(\mathbf{Z}_1) \tag{8}$$

Thus, we have a new bound that itself does not rely on the singular values computed post-activation:

$$\sigma_i(\mathbf{D}_\alpha \odot \mathbf{Z}) < \epsilon \min\{c_1(\mathbf{D}_\alpha), r_1(\mathbf{D}_\alpha)\}\sigma_1(\mathbf{Z}) \tag{9}$$

| Dataset | # Samples | Subspace | Type |
|---------|-----------|----------|------|
| Gaussian | 16384 | $\mathbb{R}^{N \times m}$ | Numeric |
| Sinusoids | 16384 | $\mathbb{R}^{N \times m \times T}$ | Numeric |
| MNIST | $6 \times 10^4$ | $\mathbb{R}^{N \times H \times W}$ | Image |
| CIFAR10 | $6 \times 10^4$ | $\mathbb{R}^{N \times H \times W}$ | Image |
| TinyImageNet | $10^5$ | $\mathbb{R}^{N \times 3 \times H \times W}$ | Image |
| WikiText | $> 10^8$ | $\mathbb{R}^{N \times |V| \times T}$ | Text |
| Multi30k | $> 3 \times 10^5$ | $\mathbb{R}^{N \times |V| \times T}$ | Text |

Table 1: The Data Sets evaluated as part of our empirical verification. For space, only the Gaussian and Sinusoid data sets are included in the main body of text, and the remaining data sets are included in the supplementary material.

| Model | Datasets |
|-------|----------|
| MLP | Gaussian |
| Elman RNN | Sinusoids |
| BERT | WikiText |
| XLM | Multi30k |
| ResNet16 | MNIST, CIFAR10, ImageNet |
| VGG11 | MNIST, CIFAR10, ImageNet |

Table 2: The models evaluated as part of our empirical verification. For space and demonstrating key features, we include results from the Fully Connected Network, Elman RNN, and ResNet16 in the main text. Additional model architectures are included in the supplement.

Because we are dealing with Leaky-ReLU activations in particular, the 2-norm of $c_i(\mathbf{D}_\alpha)$ and $r_i(\mathbf{D}_\alpha)$ take on a particular closed form. Indeed, we have

$$c_i(\mathbf{D}_\alpha) = \sqrt{N_- \alpha^2 + N_+} \qquad\qquad r_i(\mathbf{D}_\alpha) = \sqrt{M_- \alpha^2 + M_+}$$

where $N_-$ is the number of rows in column $i$ which fall into the negative domain of the Leaky-ReLU activation, and $N_+$ is the number of rows which fall into the positive domain. Similarly, $M_-$ and $M_+$ count the number of columns in the negative and positive domains in row $i$. In other words, we can exactly determine the threshold at which a singular value will fail to contribute to numerical estimation of rank if we know the non-negative coefficient and can count the number of samples which reside in particular regions of the activation function's domain.

We note that while this bound is looser than that computed using the singular values post-activation, it does provide us with an analytical description of how that bound changes in the worst case entirely in terms of the underlying linear activation and where it falls within the domain, and the linearity parameter $\alpha$. Indeed, we see that the bound is proportional to the linearity of the slope in the negative domain and will be smallest when $\alpha = 0$. We can also see that the bound will be higher when there are mostly positive samples and features in a given batch and lower whenever samples or features fall into the negative domain.

So far, we have confined our theoretical analysis to the affect of rank on activations, and have not given any discussion to the effect on the adjoint variables $\Delta$. It turns out that a similar kind of analysis to what is performed here can be applied to *any piecewise linear function*, and since the derivative of Leaky-ReLU is piecewise-linear, we can derive a similar bound to the one provided in (9). We have provided the full derivation for the affect of Leaky-ReLUs on adjoint rank in the supplement.

## 4 Empirical Methods

In this section, we describe how we perform empirical verification and practical demonstration of our theoretical results from the previous section. Since our focus is on gradient rank enforced by architectural design and not how rank may change dynamically during training, we omit many of the standard model-training results such as loss curves or acuraccy metrics from the main text; however, these figures are included in the supplementary material for this work.

We perform two broad classes of experiment: simple verification, and large-scale demonstration. In the first class of experiments, we show how the bounds derived in our theoretical results appear in simple, numerical experiments, where it is easy to verify how particular architectural choices and level of nonlinearity affect the bound of the gradient rank. In the second class of experiments, we perform larger-scale experiments and demonstrate how our derived bounds can also affect these models in practice. These latter class of experiments utilize models which include modules such as Drop-Out, BatchNorms, and LayerNorms. We do not explore these additional modules theoretically in this work, even though they may have an influence on

gradient rank [Daneshmand et al., 2020]; however, we believe this allows for a productive direction for future theoretical and empirical work.

In both styles of experiment, the primary architectural elements we demonstrate when possible are: 1) bottleneck layers, i.e., layers within a network which have a much smaller number of neurons than their input and output spaces, 2) length of sequences in parameter tying, 3) low-rank input/output spaces, 4) level of non-linearity in hidden activations.

For the sake of numerical verification, we implement auto-encoding networks on two numerical datasets. For our first data set, we generate an $m$-dimensional gaussian $X \in \mathbb{R}^{N \times m}$ as $\mathbf{x} \sim \mathcal{N}(\mu_i, \Sigma_i)$ We then use a fully-connected network as an auto-encoder to reconstruct the random samples.

Our second kind of numerical data is generated by sine waves of the form $\mathbf{x}_{i,j} = a_{i,j} \sin(b_{i,j}t) + c_{i,j}$, where we sample the parameters $a_{i,j}, b_{i,j}, c_{i,j}$ for a particular sample $i$ and dimension $j$ independently from their own univariate Gaussian distributions $a_{i,j} \sim \mathcal{N}(\mu_a, \sigma_a), b_{i,j} \sim \mathcal{N}(\mu_b, \sigma_b), c_{i,j} \sim \mathcal{N}(\mu_c, \sigma_c)$. We choose $t$ to be $T$-many points in the interval $[-2\pi, 2\pi]$. We then used an RNN with Elman Cells, GRUs and LSTMs to reconstruct the input sequence, and demonstrate how four architectural principles affect our derived bound gradient rank.

For our larger-scale experiments, we choose two popular data sets from computer vision and natural language processing. We choose Cifar10 [Krizhevsky et al., 2009] and a Tiny-ImageNet (a Subset of ImageNet [Deng et al., 2009]) for computer vision, and we choose WikiText [Merity, 2016] for natural language processing. Because our empirical analysis requires repeated singular value decompositions and multiple architectural tweaks, we avoid overly long experiment runtimes by using relatively smaller-sized versions of popular network architectures which can fit alongside batches on single GPUs. In terms of computer-vision architectures, we utilize ResNet16 [He et al., 2016] and VGG11 [Simonyan & Zisserman, 2014], and for natural language processing, we utilize the BERT [Devlin et al., 2018] and XLM [Lample & Conneau, 2019] transformers for Language Modeling and Machine Translation respectively. Again, since we are interested primarily in rank as initially bounded by architecture, we do not use pretrained weights, and we only train models for a maximum of 100 epochs.

For both numerical verification and large-scale experiments, all models were trained using $k = 5$-fold cross validation, with 4 uniquely initialized models on each fold for a total of 20 models per result. The rank metrics in the following section are thus accumulated over these 20 runs.

## 5 Empirical Results

In this section, we present some of the key results of our empirical verification and large-scale demonstrations. As the main results in this work are theoretical, we only include a few selected empirical results to demonstrate a few key empirical hypotheses which emerge naturally from out theoretical work. We include a number of additional data sets and architectures in the supplementary material which demonstrate variants or minor extensions of the hypotheses investigated here. For a complete list of which data sets and architectures are included in the main text or in the supplement see tables 1 and 2.

### 5.1 Numerical Verification

***Hypothesis 1:*** *Bottleneck layers reduce gradient rank throughout linear networks.* The bound computed in (1) suggests that reducing the size of a particular layer in a fully-connected network will reduce gradients throughout that network, regardless of the size of the input activations or adjoints computed at those layers. In Figure 1, we provide a demonstration of this phenomenon in a simple fully-connected network used for reconstructing gaussian mixture variables. In Figure 1 (left), we provide the numerical estimate of the gradient rank at each in a 3-layer network with each layer having the same dimension as the input ($d = 128$). In Figure 1 (right), we provide the same rank estimates withe the middle layer being adjusted to contain only 16 neurons. We clearly see that our bound in (1) holds, and we can see that the two layers preceding the bottleneck have their rank bounded by the adjoint, and the following layers are bounded by activation rank.

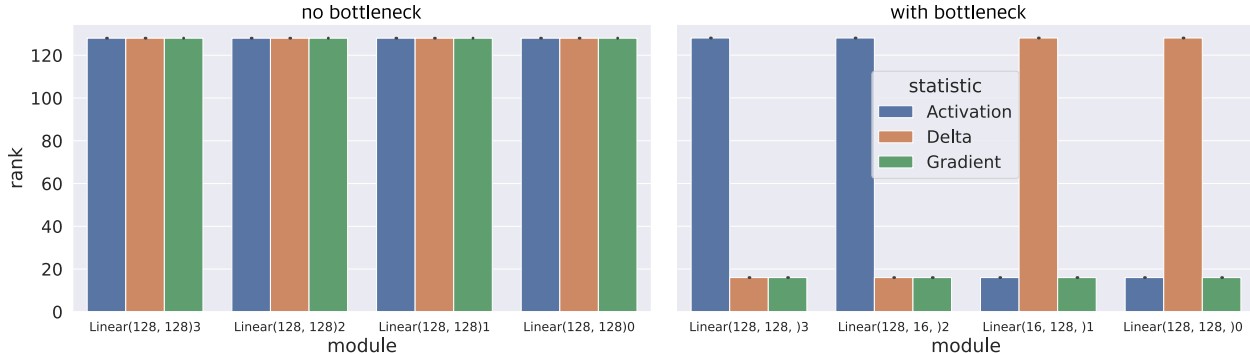

Figure 1: For a 3-layer Linear FC network, we plot the mean rank of gradients, activation, and deltas change with respect to the size of a neuron bottleneck in the middle layer. The axis axis provides the name of the module, with depth increasing from right to left. In each panel, green, blue and orange bars represent the estimated rank of gradients, activations and deltas respectively. Black vertical lines on a bar indicate the standard error in the mean estimated rank across folds and model seeds.

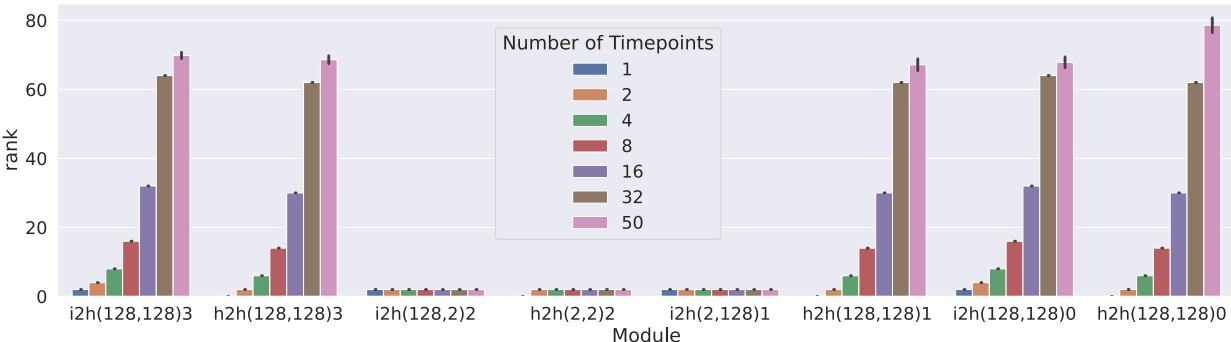

Figure 2: For a 3-layer Elman-Cell RNN, we show how mean rank of gradients, activation, and deltas change with respect to the number of timepoints used in truncated BPTT. The x axis groups particular modules, with depth increasing from right to left. Each colored bar shows the mean estimated rank over multiple seeds and folds using a different sequence length for truncated BPTT.

**Hypothesis 2:** *Parameter-Sharing such as in BPTT restores gradient rank according to the number of points in the accumulated sequence* In §3.2 we discussed how parameter-tying restores gradient rank in models which accumulate gradients over sequence, such as RNNs using BPTT (§3.2.1) or CNNs (§3.2.2) accumulating over an image. Indeed, the number of points over which back-propagation is performed will affect how much of the gradient rank is restored in the presence of a bottleneck. In Figure 2, we demonstrate the gradient rank in an 3-layer Elman-Cell RNN [Elman, 1990] trained to reconstruct high-dimensional, sinusoidal signals. We introduce a severe bottleneck in the second RNN, constraining its hidden units to 2, with the other RNNs having 128 hidden units each. We demonstrate how the introduced gradient bottleneck is ameliorated in the adjacent layers according to the number of timepoints included in truncated BPTT over the sequence. With a maximum of 50 timepoints, the bottleneck can at most be restored to a rank of 100. We show how this phenomenon extends to image size in larger scale ResNet and VGG11 networks in figures 6 and 7 in the supplement, and in BPTT in a BERT transformer in 8.

**Hypothesis 3:** *Using the derivation in §3.3, we can compute the bound over which an estimated singular value will contribute to the rank, without computing the eigenvalues themselves* One of the primary empirical upshots of the theoretical work in §3.3.1 is that using only the singular values of the underlying linearity and a count of how samples contribute to particular areas of the domain of our nonlinearity, we can compute the bound over which singular values will cease to contribute to the estimated rank. In Figure 3 we construct a low-rank random variable by computing the matrix product of two low-rank gaussian variables. We then

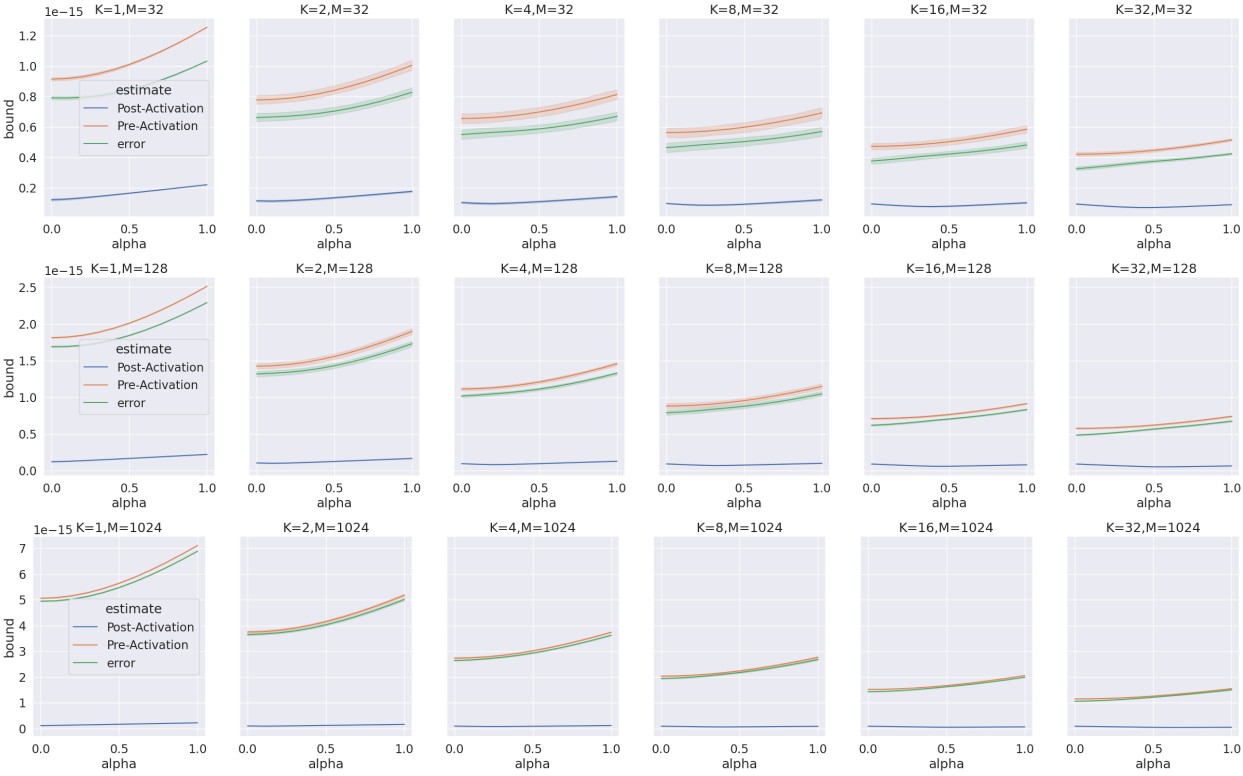

Figure 3: A numerical exploration of the derived boundary over which a given eigenvalue computed on a Leaky-ReLU activation $\sigma_k$ will cease to contribute to the rank. For each experiment we generate 1000 $M \times M$ matrices with a known latent rank $k$, and we compute the singular value bound for contribution to the rank using the singular values with the post-activation singular values (blue curve) and then the pre-activation singular values using equation 5 (orange curve). We also plot the error between the post and pre-activation bounds (green curve). (9) with a blue dotted line. For each experiment we show how the bound changes as a function of the linearity $\alpha$ of the leaky-relu activation function.

compute the estimated eigenvalues after applying a Leaky-ReLU nonlinearity. We then estimate the bound to rank contribution first by computing it using the maximum estimated singular value, and then using the boundary we derived in (9) which does not require singular values. This empirical demonstration indicates that this derived bound is equivalent to numerical boundary on singular value contribution to rank computed from the output of the leaky-ReLU. This demonstrates that our theoretical result allows for exact prediction of when singular values will no longer contribute to rank, using only the singular values of the input to the activation function. For all experiments we used 64-bit floating point numbers, and we see that our bound is vanishingly close to the actual estimate, and the corresponding rank estimates are exactly correct. We additionally performed experiments with 32 bit floating point numbers and though the accumulated error is larger between the pre and post activataion bounds, it is still lower than $1e^{-6}$ - full plots for these experiments and the rank estimates for both numerical types can be found in the supplements in figures 9, 10, 11 and 12.

**Hypothesis 4:** *The negative slope of the Leaky-ReLU activation is related to the amount of gradient rank restored.* Although our theoretical analysis in §3.3.1 was given primarily in terms of how Leaky-ReLU contributes to the rank of the input activations, it stands to reason that the resulting product of activations and adjoint variables would be affected by the negative slope as well. In Figure 4, we implement a 3-layer fully-connected network for reconstruction of Gaussian variables, and we compute the rank of the gradients changing as a function of the slope of the Leaky-ReLU activation. We use a network with 128 neurons at the input and output layers and a 2-neuron bottleneck layer. Our results indicate that pure ReLU activations do not fully restore gradient rank in either direction of back-propagation; however, the negative slope does

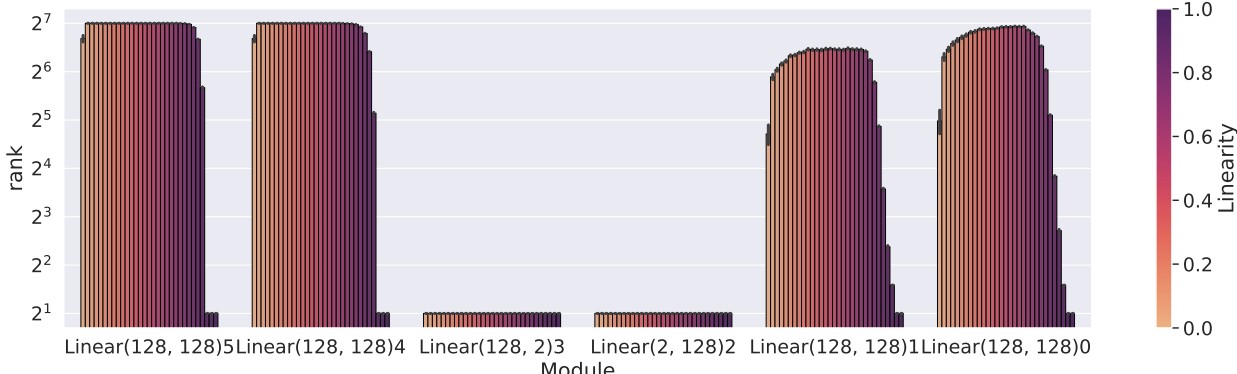

Figure 4: For a 5-layer (6 weight) FC network with Leaky-ReLU activations, we show how mean rank of gradients, activation, and deltas change with respect to the negative slope $\alpha$ of the noninearity. Layer sizes are plotted on the x axis with the depth increasing from left to right. We enforce a bottleneck of 2 neurons in the central layer. For each module, we estimate the rank and provide a colorbar corresponding to the level of nonlinearity increasing in the range of [0,1].

increase estimated rank close to full-linearity. As the activations approach linearity, the rank returns to the gradient bottleneck.

## 6   Discussion

Our theoretical bound on gradient rank in linear networks provides a number of startling insights. As we empirically demonstrate in Figure 1, linear networks with bottleneck layers are constrained to low-rank learning dynamics, where the rank cannot exceed that of the smallest number of neurons in the network. Beyond the somewhat artificial introduction of bottlenecked layers, these constraints also emerge when dealing with low-dimensional input spaces (even when they are embedded in higher-dimensional spaces like in Figure 1 in supplement). Perhaps more startling is the realization that in supervised learning tasks with relatively few output variables (such as only 10 neurons for 10 classes in Cifar10/MNIST, which can be seen in the rank of the linear classifier in Figures 2+3), the gradient rank throughout the entire network will not exceed that number of variables. When using linear activations, this finding suggests that the ramifications of neural collapse to input layers beyond the terminal linear classifier, with Neural Collapse playing a role at *all* phases of training, not just during the terminal phase. Since our analysis in this work is on architectural constraints which will affect dynamics during *all* training, further work studying the actual gradient dynamics and their effect on the weights during training is needed to fully flesh-out the connection to neural collapse.

Through our analysis of linear networks, we also provide an explanation for how parameter-tying mitigates the tight constraints on rank which may emerge due to bottlenecks. In Figures 3 and 4 our empirical verification demonstrates that when low-rank gradients computed at each point in the sequence, rank may be partially restored; however, the level to which rank is restored depends on the length of the sequence. The implication for networks with parameter tying is that aggressively truncated sequences in parameter-tied networks will still be affected by low-rank gradient dynamics.

Our theoretical result identifying how to compute the numerical boundary on rank for Leaky-ReLU networks provides a novel theoretical extension to how nonlinear activations can affect learning dynamics. Additionally, the ability to control the negative slope of the Leaky-ReLU activations allows us to demonstrate how numerical precision can affect the bounds. At the end of our analysis; however, we are left with a boundary that is highly data-dependent, and as we show in Figure 4 even fully nonlinear ReLU activations may lower the numerical estimation of rank. This remaining data-dependence in our theory suggests that there is only so much insight to be gained from architectural choice alone, and future work will require analysis of how particular input and output spaces may impose particular boundaries or dynamics on gradient rank. This input/output analysis may also provide deeper insights and tighter bounds the affects of nonlinear activations

and parameter tying in particular, with highly correlated or sparse input and output spaces potentially affecting bounds and dynamics.

One limitation of our analysis in this work is that we have purposely avoided relating the emergence of low-rank phenomenon to model performance on particular tasks. Our reason for shying away from this discussion is that model performance is closely related to dynamics throughout the entire training phase, and our theoretical results apply to networks at *any* phase of training, and as such are agnostic to whether a model performs well or not. Our work here provides ample groundwork for analyzing the dynamics within our derived boundaries, and so we leave the connection of gradient rank collapse and performance as future work which can focus on correspondences between collapse and dynamics.

A further limitation of our analysis is that we have confined the work to constraints on how gradient rank without taking into account how rank may change during training. This step is needed in order to fully realize the connection to Neural Collapse and to the kinds of dynamics observed by Saxe et al. ;however, our analysis has shown that *regardless of dynamics* we can provide sensible contraints on gradient rank entirely on the architecture alone. This step in effect sets the stage for further studies of dynamics in terms of the rank or spectrum of the gradients during learning by defining the boundaries within which those dynamics can occur.

An additional constraint of our analysis here is our restriction to analyzing linear and Leaky-ReLU networks. Our primary reason for doing this is that with minimal theoretical pretext, Linear and Leaky-ReLU networks call be discussed within the machinery of linear algebra. Other nonlinear activations such as the hyperbolic tangent, logistic sigmoid, and swish require significant theoretical pretext before the linear-algebraic notions of singular values and rank can be applied. Future work may be able to draw on mathematical fields of functional analysis (in the style of Dittmer et al. [2019]), algebraic topology [Cox et al., 2005], or analysis of linearized activations (as is done in Bawa & Kumar [2019]) to provide theoretical frameworks for studying low-rank dynamics. We leave this as a substantial opening for future work.

### 6.1 Connection to ReLU-Singular Values

One initial theoretical answer to analyse particular nonlinearities involves extending the notion of singular values to certain nonlinear functions which are locally linear, such as ReLU or Leaky-ReLU activations [Dittmer et al., 2019]. For the nonlinear operator $\phi_\alpha(\mathbf{Z}) = \text{LeakyReLU}_\alpha(\mathbf{Z})$, where $\alpha \in [0,1]$ controls the slope of the activation when $\mathbf{Z} < 0$. We note that when $\alpha = 0$, this is equivalent to a ReLU activation and when $\alpha = 1$ it is equivalent to a Linear activation.

Following Dittmer et al. [2019], for a matrix $\mathbf{Z} \in \Re^{m \times n}$ the $k$th "Leaky-ReLU Singular Value" is defined for the operator $\phi_\alpha(\mathbf{Z})$ as

$$s_k(\phi_\alpha(\mathbf{Z})) = \min_{\text{rank } \mathbf{L} \leq k} \max_{x \in \mathcal{B}} \|\text{LeakyReLU}_\alpha(\mathbf{Z}x) - \text{LeakyReLU}_\alpha(\mathbf{L}x)\|. \tag{10}$$

In Dittmer et al. [2019], the extension of the notion of ReLU singular values to Leaky-ReLUs carries naturally; however, for completeness, we have included Leaky-ReLU-specific versions of each of the proofs from that work in the supplement. Among these results, we have the following lemma:

*Lemma:* Let $\phi_\alpha(\mathbf{Z}) = \text{LeakyReLU}_\alpha(\mathbf{Z})$ for $\mathbf{Z} \in \Re^{n \times m}$, then

$$s_k(\phi_\alpha(\mathbf{Z})) \leq \sigma_k(\mathbf{Z}) \tag{11}$$

In other words, the Leaky-ReLU singular values will be bounded above by the singular values of the underlying linear transformation $\mathbf{Z}$. It follows then that as we increase $\alpha$ along the interval [0, 1], the analogous notion of " rank (the number of non-zero values of $s_k$) will converge converge upward to the linear rank of $\mathbf{Z}$, and our boundaries will still hold.

## 7 Conclusion

In this work, we have presented a theoretical analysis of gradient rank in linear and Leaky-ReLU networks. Specifically, we have shown that intuitive bounds on gradient rank emerge as direct consequences of a number

of architectural choices such as bottleneck layers, parameter tying, and level of linearity in hidden activations. Our empirical verification and demonstration illustrate the computed bounds in action on numerical and real data sets. The bounds we provide connect simple principles of architectural design with gradient dynamics, carving out the possible space in which those dynamics may emerge. Our work thus serves as a groundwork for continued study of the dynamics of neural spectrum collapse and gradient dynamics, not only in linear classifiers, but in many classes of network architecture.

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

## A  Notation

See Table 3.

| Symbol | Subspace | Definition |
|:---:|:---:|:---|
| *Data Set Notation* | | |
| $m$ | $\mathbb{N}_+$ | Number of Input Features |
| $\mathbf{x}$ | $\Re^m$ | Single input Sample |
| $N$ | $\mathbb{N}_+$ | Number of Samples |
| $\mathbf{X}$ | $\Re^{N \times m}$ | Data set features with $N$ samples and $m$ features |
| $n$ | $\mathbb{N}_+$ | Number of Target Features |
| $\mathbf{y}$ | $\Re^n$ | Single target sample |
| $\mathbf{Y}$ | $\Re^{N \times n}$ | Data set targets with $N$ samples and $n$ features |
| $T$ | $\mathbb{N}_+$ | Length of Data Sequence |
| $\mathcal{X}$ | $\Re^{N \times m \times T}$ | Data set feature tensor: $N$ samples, $m$ features, sequence length $T$ |
| *Network Architecture* | | |
| $L$ | $\mathbb{N}_+$ | Number of Layers in Neural Network |
| $h_i$ | $\mathbb{N}_+$ | Number of Neurons at Network Layer $i$ |
| $\mathbf{W}_i$ | $\Re^{h_{i-1} \times h_i}$ | Weight matrix from layers $i-1$ to $i$ |
| $\mathbf{b}_i$ | $\Re^{h_i}$ | Bias at layer $i$ |
| $\phi_i$ | $\phi_i : \Re^{h_i} \to \Re^{h_i}$ | Activation function at layer $i$ |
| $\mathbf{\Phi}(\{\mathbf{W}_i\}_{i=0}^L, \{\mathbf{b}_i\}_{i=0}^L, \{\phi_i\}_{i=0}^L)$ | $\mathbf{\Phi} : \Re^m \to \Re^n$ | $L$-layer Neural network |
| $h_0 = m$ | $\mathbb{N}_+$ | Number of Neurons at Input |

Table 3: Important notation used throughout the main work.

# B   Parameter Tying: Bounds on Gradients in Convolutional Layers with Linear Activations

Our derivation of bounds on the gradient rank of convolutional layers will follow much of what was derived for RNNs. The primary difference will appear in the number of steps over which gradients are accumulated, and their relationship to image size, stride, kernel size, padding, etc.

Suppose we are working on a convolution of dimension $m$. If we let $N$ denote the size of a given batch, and let $C_{in}, C_{out}$ be the input/output channels, and let $\mathbf{w}_{in} \in \mathbb{N}_+^m$ be a list of image dimensions (such as Height and Width for a 2d image). Then the input image to the convolution is denoted as a tensor $\mathcal{X} \in \Re^{N \times C_{in} \times w_{in,1} \times w_{in,2} \times \cdots \times w_{in,m}}$.

Suppose we are performing a convolution with a kernel sizes $\mathbf{k} \in \mathbb{N}_+^m$, dilations $\mathbf{d} \in \mathbb{N}_+^m$, padding $\mathbf{p} \in \mathbb{N}_+^m$, and stride $\mathbf{s} \in \mathbb{N}_+^m$. The weight tensor for this convolution is denoted as $\mathcal{W} \in \Re^{C_{out} \times C_{in} \times \mathbf{k}_1 \times \mathbf{k}_2 \times \cdots \times \mathbf{k}_m}$. The output of this convolution is thus computed as the tensor $\mathcal{Y} \in \Re^{N \times C_{out} \times w_{out,1} \times w_{out,2} \times \cdots \times w_{out,m}}$:

$$\mathcal{Y}_{i,j,\ldots} = \sum_{k=1}^{C_{in}} \mathcal{W}_{i,k,\ldots} \star \mathcal{X}_{i,k,\ldots}$$

where $\star$ is the $m$-dimensional cross-correlation operator.

Following reverse-mode auto-differentiation for $m$-dimensional convolutions, the gradient of $\mathcal{W}$ is computed as a convolution between the input $\mathcal{X}$ and the adjoint from the backward pass $\boldsymbol{\Delta}$:

$$\nabla_{\mathcal{W}} = \sum_{i=1}^{N}\{\mathcal{X} \star \boldsymbol{\Delta}\}_i$$

It is clear that with linear activations, even if $\mathcal{X}$ and $\boldsymbol{\Delta}$ are low-rank, the rank of this gradient is accumulated over the cross-correlation operator $\star$ (as well as over $N$, in a similar way to for linear layers). It becomes apparent that like for RNNs, we are accumulating over the sequence of $\prod_{i=1}^{d} w_{out,i}$ many patches, where $w_{out,i}$ is the size of the output in the $i$th dimension.

For convolutional layers, this can be explicitly computed as

$$w_{out,i} = \left\lfloor \frac{w_{in,i} + 2p_i - d_i \times (k_i - 1) - 1}{s_i} + 1 \right\rfloor$$

if we let $\mathcal{B}$ be the linear bound computed in section 2.3, then the bound on the rank of the gradient is thus

$$\text{rank}(\nabla_{\mathcal{W}_i}) \le \mathcal{B} \prod_{i=1}^{m} \left\lfloor \frac{w_{in,i} + 2p_i - d_i \times (k_i - 1) - 1}{s_i} + 1 \right\rfloor \tag{12}$$

Intuitively, this means the bound will shrink as the input image size and padding shrinks, and will shrink as the stride, dilation and kernel size increase.

## C  Rank of the Leaky-ReLU Derivative

We can use a similar kind of analysis as in section 3.3.1 to derive a bound on the rank for the partial derivative of the Leaky-ReLU activation on the output. As we mentioned in that section, the analysis can extend trivially to any piecewise linear function, and indeed the derivative of a Leaky-ReLU is piecewise linear.

Let $\mathcal{D}_\alpha(\mathbf{Z}) \in \Re^{h \times h}$ be the matrix with entries corresponding to the linear coefficients from the Leaky-Relu activation applied to internal activation $\mathbf{Z}_i = \mathbf{X}\mathbf{W}_i$. The partial derivative w.r.t to the output can be written as the hadamard product

$$\phi_\alpha'(\mathbf{Z}_i) = \mathbf{D}_\alpha \odot \mathbf{1}^{h \times h} \tag{13}$$

where $\mathbf{1}^{h \times h}$ is an $h \times h$ matrix of ones.

From our derivation in section 3.3.1, we can say that $\mathbf{D}_\alpha \odot \mathbf{1}^{h \times h}$ will remain numerically rank-deficient according to the bound

$$\min\{c_k(\mathbf{D}_\alpha), r_k(\mathbf{D}_\alpha)\} \le \epsilon \min\{c_1(\mathbf{D}_\alpha), r_1(\mathbf{D}_\alpha)\}\sigma_1(\mathbf{Z}_i)/\sigma_k(\mathbf{Z}_i) \tag{14}$$

We note that $\mathbf{1}^{h \times h}$ is rank-1 with $\sigma_1 = h$ and $\sigma_k \le \epsilon h$. If we suppose this bound is saturated, the bound depends primarily on the quantity $\min\{c_1(\mathbf{D}_\alpha), r_1(\mathbf{D}_\alpha)\}$, and will loosen somewhat as the precision of $\sigma_k$ shrinks it toward 0.

## D  Additional Empirical Verification

***Hypothesis S1***: *Low-rank input/output spaces systematically bound gradient rank.* The remaining terms in the bound in (12) show that gradient rank is bound by the rank of the inputs and the rank partial derivative

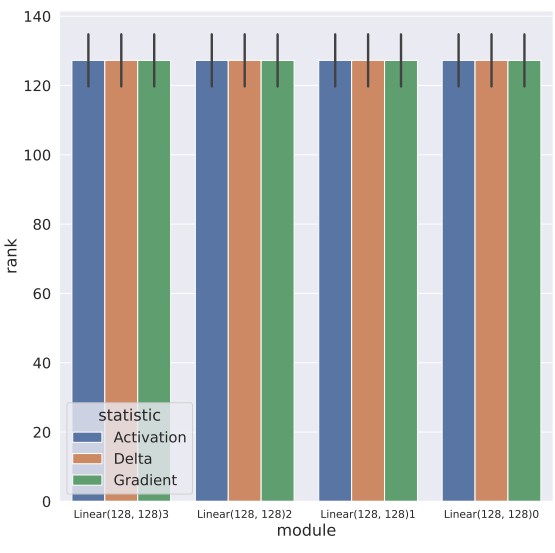

(a) Full-dimensional input

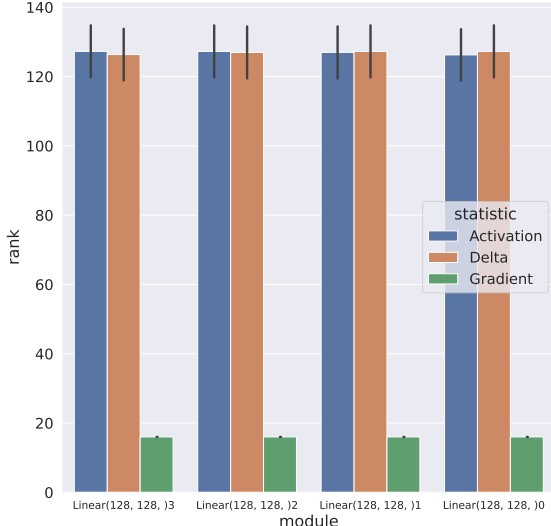

(b) Input of dimension 16 embedded

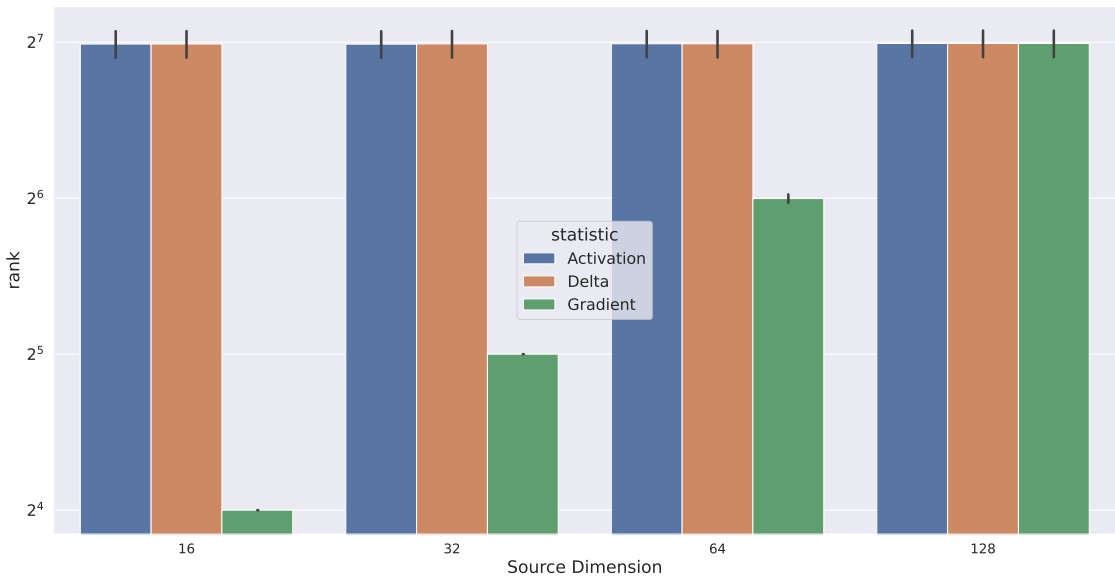

(c) Gradients, Deltas, Activations for different dimension input

Figure 5: Low-Dimensional Input

of the loss function (i.e., the rank of the output space). In Figure 5, we show the computed gradient rank on a fully-connected network with three layers of 128 neurons each, trained to reconstruct a full-rank (128-dim) gaussian input (panel 5a) in contrast to a model trained to reconstruct a low-rank (16-dim) gaussian embedded in a higher-dimensional (128-dim) space (panel 5b). Indeed, in panel 5c we see that any linear model which receives the low-rank embedded input has gradients which are bounded entirely by the rank of the input space.

# E   Large Scale Demonstration

In this section, we include results on larger-scale networks and problem settings. Our goal is to illustrate how architectural choices can affect rank not only in small models, but also in modern architectures. These results highlight the relevance of our theoretical result to the deep learning community at large.

***ImageNet with ResNet18:*** In Figure 6, we illustrate the effect of increasing the input image size on the rank of the gradient in the ResNet18 architecture. In each panel the image size increases from top to bottom, and as expected smaller image sizes produce smaller ranks.

Additionally, we estimated the effect of artificially introduced Leaky-ReLU activations with different levels of $\alpha$; however, we observed little noticeable effect as long as the input image was large.

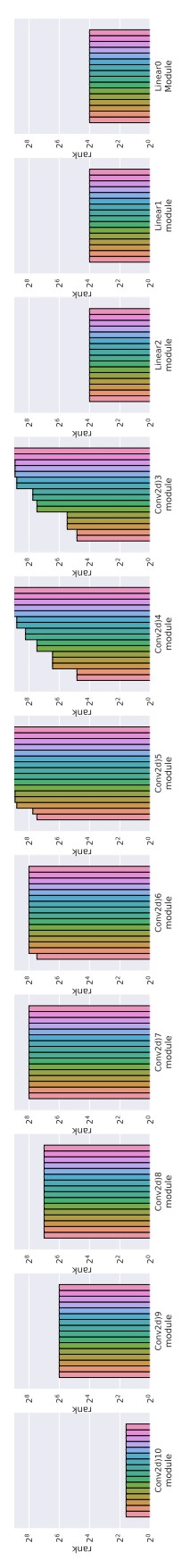

Figure 6: Illustration of the rank of the gradient at each layer in the ResNet18 architecture used to classify Tiny-Imagenet. Each panel shows the effect of increasing image size (from top to bottom) on the rank, illustrating that larger image spaces provide more accumulation of the gradient.

Figure 7: Illustration of the rank of the gradient at each layer in the VGG11 architecture used to classify Tiny-Imagenet. Each panel shows the effect of increasing image size (input is from top to bottom) on the rank, illustrating that larger image spaces provide more accumulation of the gradient.

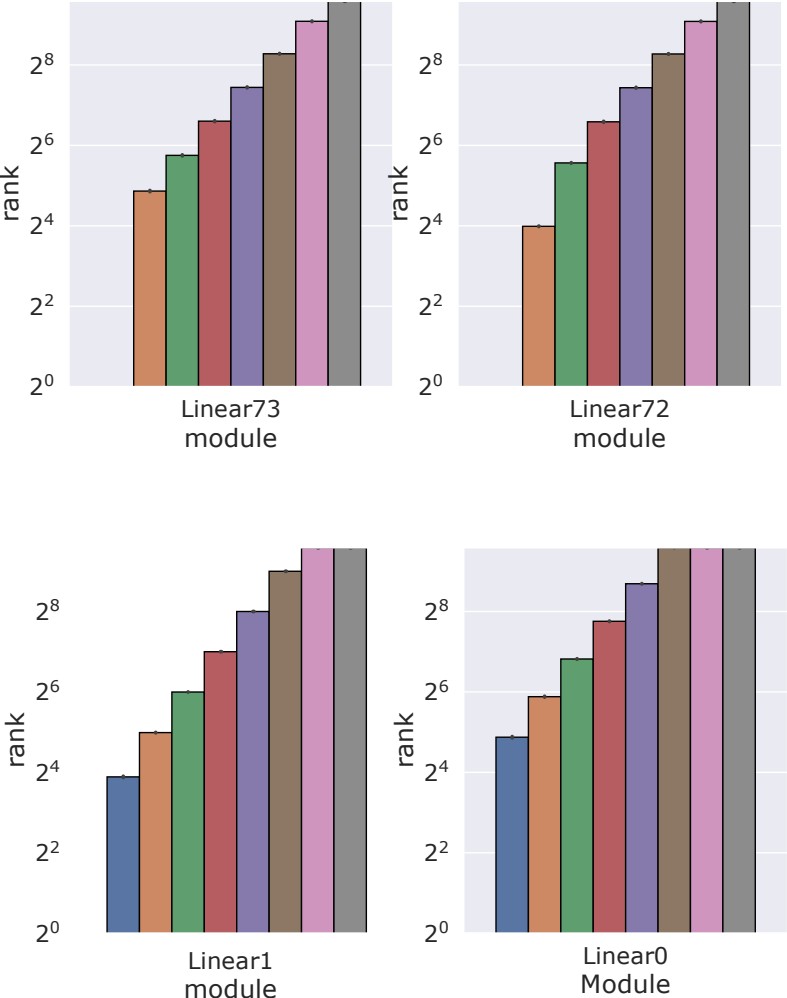

Figure 8: Illustration of the rank of the gradient at each layer in the BERT architecture used for language modeling on the WikiText2 data set. Each panel shows the effect of increasing sequence length (increasing from right to left) on the rank, illustrating that larger sequence lengths. provide more accumulation of the gradient.

***ImageNet with VGG11:*** In Figure 7 we illustrate the effect of increasing the input image size on the rank of the gradient in the ResNet18 architecture. In each panel the image size increases from top to bottom, and as expected smaller image sizes produce smaller ranks.

***WikiText/M130k with BERT/XLM:*** We have included a demonstration of decreasing the sequence length in two large-scale NLP datasets (WikiText and Multi30K) for the BERT architecture with standard pretraining and XLM pretraining. In Figure 8 we include the estimated rank of the first two and last two linear layers for BERT applied to the WikiText data set. The rank for all other layers and for the4 XLM/Multi30k application are included in separate PDFs along with this supplement. In general we do not see much variation between linear layers in each architecture, except for near the input (layers 73,72), in which length 1 sequences collapse the rank of the gradients down to 1.

## F  Full Figures for Numerircal Bound Verification

In this section, we include the full experimental results for the exploration of our pre-activation bound on the rank as we showed partially in figure 3. Using 32-bit floating point numbers, figure 9 shows the estimated

bound under which singular values no longer contribute to the rank computed using the post-activation singular values and the theoretical boundary we provide in equation 9. Figure 10 shows the corresponding rank estimation using each of the bounds. Figures 11 and 12 provide the same experiment using 64-bit floating point numbers.

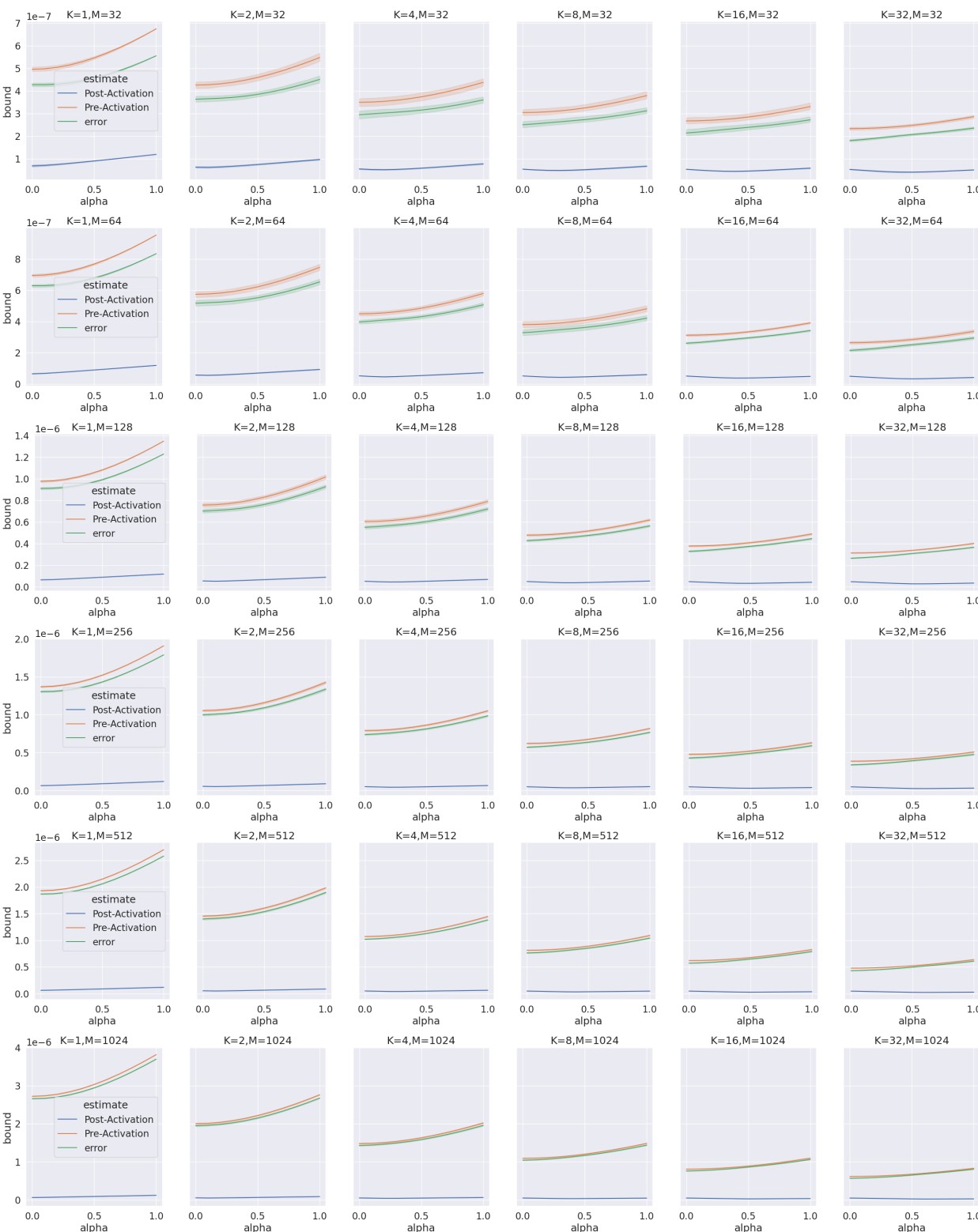

Figure 9: Using 32-Bit Floating Point Numbers: A numerical exploration of the derived boundary over which a given eigenvalue computed on a Leaky-ReLU activation $\sigma_k$ will cease to contribute to the rank. For each experiment we generate 1000 $M \times M$ matrices with a known latent rank $k$, and we compute the singular value bound for contribution to the rank using the singular values with the post-activation singular values (blue curve) and then the pre-activation singular values using equation 5 (orange curve). We also plot the error between the post and pre-activation bounds (green curve). (9) with a blue dotted line. For each experiment we show how the bound changes as a function of the linearity $\alpha$ of the leaky-relu activation function.

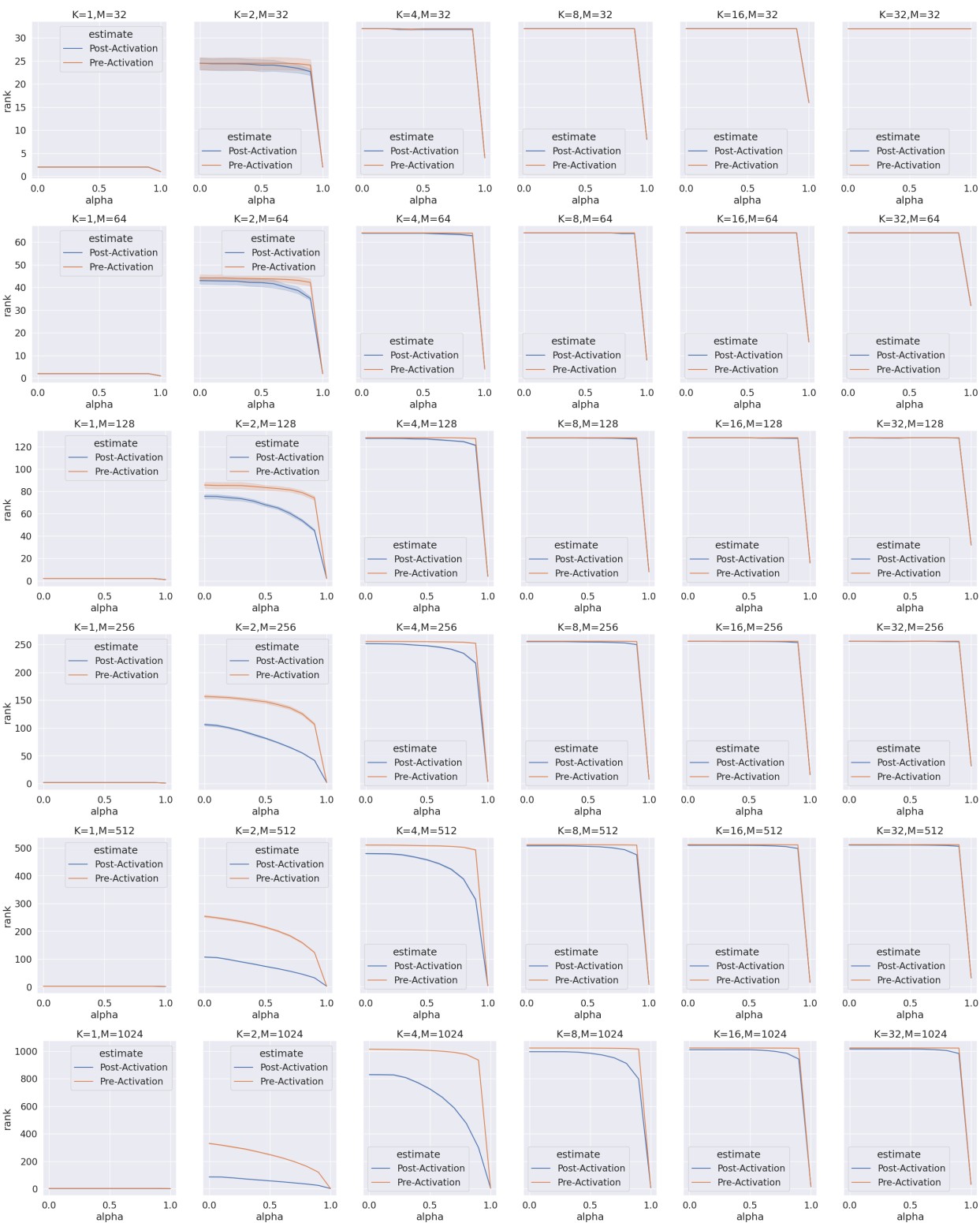

Figure 10: Using 32-Bit Floating Point Numbers: A numerical exploration of the derived rank using the post and pre-activation boundaries. For each experiment we generate 1000 $M \times M$ matrices with a known latent rank $k$, and we compute the singular value bound for contribution to the rank using the singular values with the post-activation singular values (blue curve) and then the pre-activation singular values using equation 5 (orange curve). We also plot the error between the post and pre-activation bounds (green curve). (9) with a blue dotted line. For each experiment we show how the bound changes as a function of the linearity $\alpha$ of the leaky-relu activation function.

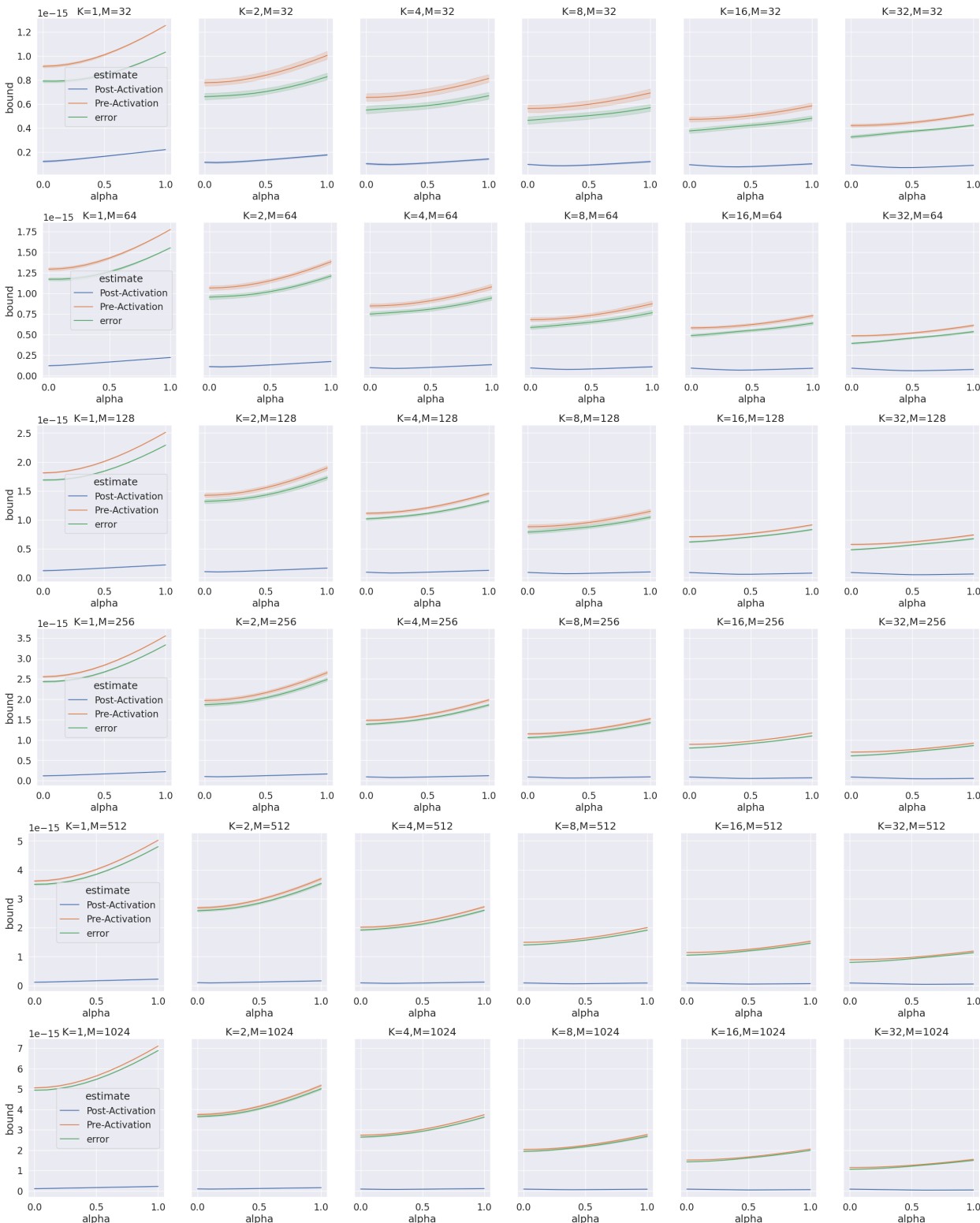

Figure 11: Using 64-Bit Floating Point Numbers: A numerical exploration of the derived boundary over which a given eigenvalue computed on a Leaky-ReLU activation $\sigma_k$ will cease to contribute to the rank. For each experiment we generate 1000 $M \times M$ matrices with a known latent rank $k$, and we compute the singular value bound for contribution to the rank using the singular values with the post-activation singular values (blue curve) and then the pre-activation singular values using equation 5 (orange curve). We also plot the error between the post and pre-activation bounds (green curve). (9) with a blue dotted line. For each experiment we show how the bound changes as a function of the linearity $\alpha$ of the leaky-relu activation function.

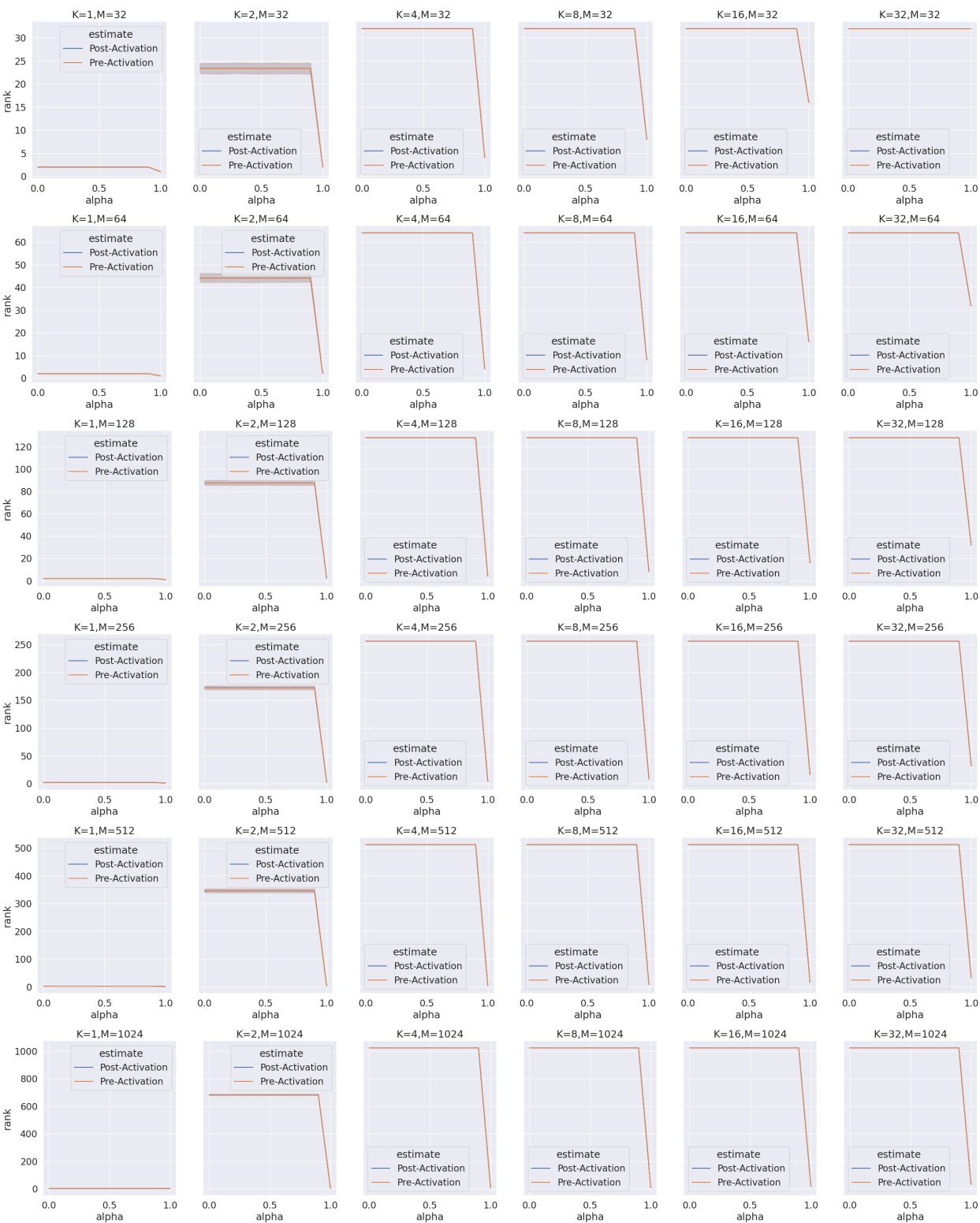

Figure 12: Using 64-Bit Floating Point Numbers: A numerical exploration of the derived rank using the post and pre-activation boundaries. For each experiment we generate 1000 $M \times M$ matrices with a known latent rank $k$, and we compute the singular value bound for contribution to the rank using the singular values with the post-activation singular values (blue curve) and then the pre-activation singular values using equation 5 (orange curve). We also plot the error between the post and pre-activation bounds (green curve). (9) with a blue dotted line. For each experiment we show how the bound changes as a function of the linearity $\alpha$ of the leaky-relu activation function.

