# OpenReview forum: "Low-Rank Learning by Design: the Role of Network Architecture and Activation Linearity in Gradient Rank Collapse"
_TMLR — Rejected by TMLR_

### Review · Reviewer_o3M5 · 2023-12-15

**Summary Of Contributions:**

The authors provide theoretical analysis to the ranks of neural networks' gradients, under different assumptions like parameter sharing and the type of activation function. Empirical observations are provided to verify some aspects of the theoretical results.

**Audience:**

Yes

**Broader Impact Concerns:**

This work is more on theory side so there shall be no ethical concerns.

**Claims And Evidence:**

Yes

**Requested Changes:**

See above - some points are not clear, and new experiments might be necessary.

**Strengths And Weaknesses:**

Strengths:
- The problem studied by the authors has values as it has connection to neural collapse.
- The amount of experiments conducted by the authors is large

Weakness:

- Some results seem to be straight-forward, such as the bounds derived in Section 3.1: of course the ranks of the gradients are limited by the shape of weight matrices.
- Section 3.2 requires a lot of effort to read and can be improved:
  - What is the size of $X_t$ and $X_{t,i}$. These tensors and not previously defined, which makes me hard to understand how to derive $\nabla_U$ and $\nabla_V$, and ultimately makes me hard to understand how the full-rank are achieved.
  - It would be better to give an example to show how the full rank is achieved for broader audience.
- Section 3.3.1: What is the meaning of $i$ in Equation (5)? Is it an arbitrary index, or just $k$ inferred from the context? I'd be surprised if it is a random index because when $i=1$ this seems to be hard to hold.
- Section 3.3.1: If I understand correctly, the section is to show that when there is such a $k$, what would be the conditions it satisfy. So I'm a little bit confused since I thought I would have a bound of the value of $k$.
- Section 5:
  - As aforementioned, it would be better if the authors can provide example where the gradients can easily achieve full ranks - Figure 2 fails to show this because we see the rank is high only when T is greater than 32 (or 16 arguably). Otherwise I don't see a reason to support that argument.
  - Hypothesis 3:
    - The right side of Equation (5) is definitely small because there is an epsilon there. I think to show the efficacy of this bound, one must show the corresponding singular values of different $k$ and demonstrate that the bounds are correct.
- Minors:
  - Wouldn't the results change if the architectures have residual connections, or other more recent structures like attention modules? It would be better if the authors can provide some discussion on these topology.
  - It would be better if the authors can clearly state which parts of theory is innovative as I believe some of them might be pre-existed (at least for the linear part it is somehow straight-forward to obtain)

---

> ### Author Response · Authors · 2024-01-07
> **Thank you for your review!**
>
> We thank you for your review, and for recognizing the value of our work as well as the thorough empirical work we have done to
> bolster it! We would like to point out our general response w.r.t. the raised concern about straightforwardness, and we would like to address your concerns individually in the following comments.

---

> ### Author Response · Authors · 2024-01-07
> **Against Apparent Straightforwardness**
>
> As we have pointed out in our general response, the provided rank **is not determined only by the shape of the weight matrices**. Indeed, the **rank** (not just the shape) of the input and the rank of the loss function (particularly the partial derivative of that loss function w.r.t. the hidden activations at the output layer) also determine the rank of the gradient. Additionally, the batch size used to compute a particular gradient will also determine its rank.
>
> Although our initial theoretical arguments do use elementary linear
> algebraic identities, it is the perspective from auto-differentiation
> that provides further value. Particularly, the fact that rank
> bottlenecks flow *backwards* through the network due to the
> computation of the adjoint matrices $\mathbf{\Delta}$ elevates the argument
> to the point of novel insights such as the number of classes
> determining the rank **in the entire network**. The significance of this observation can be appreciated when viewed as evidence of the advantages of multitask learning. We firmly believe that the complexity of an argument alone does not determine its value; indeed, our perspective from auto-differentiation allows for novel insights despite the use of elementary arguments along the way.

---

> ### Author Response · Authors · 2024-01-07
> **Clarity in section 3.2**
>
> At the beginning of section 3.2.1 we provide a definition of the full input tensor $\mathcal{X} \in \mathbb{R}^{N\times n \times T}$. We omitted further definitions of particular slices of this tensor, believing it would be clear; however, we recognize it is better to provide more definitions rather than less, and we thank you for recommending this improvement!

---

> ### Author Response · Authors · 2024-01-07
> **Clarity in Section 3.3.1**
>
> In equation (4), $k$ is the rank of the matrix, and so $i$ is used to denote the $i$th singular value (up to a maximum $k$). We recognize that this section needs some rehashing of the notation to clarify that we are particularly referring to the singular value which falls below the threshold in equation (2) to determine the rank. Indeed, we used the notation $k$ in equation (2) to denote that the $k$th singular value is the last one to determine rank.  We have endeavored to fix this notational issue in our revision.
>
> Indeed the bound in section 3.3.1 is a somewhat indirect bound on $k$ itself. As we mention in that section, when the rank $k$ is computed numerically, we determine that $k$ by finding the $k+1$th singular value which falls below the bound in equation (2). We have tried to rehash the notation in this section to make things clearer.

---

> ### Author Response · Authors · 2024-01-07
> **Section 5**
>
> We would like to direct you Figure 8 in our appendix which was originally included with the submission, in which we use a Transformer-Based architecture (which does easily achieve full rank with large T) demonstrating a similar principal to what you are asking us to provide. Simply put, it turns out that even in large, complex models, truncated back-prop and sequence length with very few timepoints will result in lower rank.
>
> Additionally, with respect to hypothesis 3- the requested experiment is exactly what we have provided in figure 3. Namely, we have shown the difference between the bound from equation (2) and our derived bound that is only determined by pre-activation singular values (equation 5). We have explicitly demonstrated how this affects the actual computation of rank in figure 10 which we have included in the appendix in our original submission.

---

> ### Author Response · Authors · 2024-01-07
> **Minor Concerns**
>
> It is true the inclusion of additional modules (batch normalization, skip connections, gates, etc.) may also affect gradient rank. We believe that an empirical (and further theoretical if posssible) survey of particular module effects on gradient rank, especially on gradient rank dynamics during training, would be novel and useful contribution to the literature; however, we view this work as a next step from our own. Particularly, an extension of the analysis of nonlinear activations to sigmoid activations, tanh, etc. is a natural next step from our analysis of leaky-ReLU activations.
>
> As far as we have found in the literature, the Auto-Differentiation perspective of systematically studying gradient rank is novel. Although some of the underlying arguments are straightforward once they are exposed to the reader, we are not aware of any work which has begun from AD on linear networks and uses this basis to extend to piecewise-linear activation functions.

---

### Review · Reviewer_Rv5s · 2023-12-18

**Summary Of Contributions:**

The paper "Low-Rank Learning by Design: the Role of Network Architecture and Activation Linearity in Gradient Rank Collapse" presents a comprehensive study of gradient rank in Deep Neural Networks (DNNs). The authors focus on understanding how architectural choices and the structure of data affect the bounds of gradient rank in various network architectures, including fully-connected, recurrent, and convolutional neural networks.

**Audience:**

No

**Claims And Evidence:**

No

**Requested Changes:**

See Weaknesses above.

**Strengths And Weaknesses:**

## Strengths

1. The analysis of how leaky ReLU slope affects rank deficiency adds useful extension of the theory to nonlinear models.

2. Empirical verification is clean and clearly demonstrates the theoretical bounds in action across different architectures.

## Weaknesses
1. The primary theoretical results for linear networks appear trivial. Since a linear network equates to matrix multiplication, it seems evident that the total rank is dictated by the minimum rank among all weight matrices. Similarly, linear RNNs and CNNs are also overly simplistic. Consequently, in my view, Sections 3.1 and 3.2 offer limited new insights to the machine learning community in understanding neural networks.

2. Although the paper claims a connection to neural collapse, I believe this link is tenuous. Neural collapse is characterized by optimizing classification losses (cross-entropy & MSE) with neural networks, leading to per-class features converging to a single point in feature space. However, this paper does not explore the collapse of features, making the purported connection unclear.

3. The significance of gradient rank to the community is unclear to me. While the author has conducted experiments on gradient rank, the relevance and importance of these findings are not compellingly conveyed. I am skeptical that a segment of TMLR's audience would find these results particularly enlightening or of interest.

---

> ### Author Response · Authors · 2024-01-07
> **Thank you for your review!**
>
> Thank you for your feedback, and for recognizing the particular
> usefulness in our extension to Leaky-ReLU activations as well as the
> clarity of our empirical verification! We recognize that it is vital
> for TMLR submissions to contain novel material which is relevant for
> its readership, and we thank you for sharing your feelings on this
> matter. We have addressed concerns regarding triviality in our general
> response above; however, we would like to restate our arguments in brief here with regard to your specific concerns.

---

> ### Author Response · Authors · 2024-01-07
> **Against Apparent Triviailty**
>
> Although it is strictly correct that "a linear network equates to
> matrix multiplication", we believe this oversimplification glosses
> over the myriad important insights provided from taking a perspective
> rooted in reverse-mode auto-differentiation. As we point out in our
> general response, the total rank is in fact not just determined by the
> minimum rank of the weight matrices, but also by the rank of the input
> and the rank of the partial derivative of the loss function w.r.t. the
> internal activations at the output layer. Furthermore, the fact that a
> bottleneck affects rank in the reverse direction due to the computation of the adjoint matrices $\mathbf{\Delta}$ provides a more novel insight than a mere reliance on the matrix multiplication which appears in the forward pass alone. To take the example of classification, the insight provided by our work that the rank of the gradients **in the entire network** will be determined by the number of classes does not seem entirely obvious unless we recognize how adjoint variables play a role in determining gradient rank. Finally, the insight provided by our perspective from AD that batch size (the $N$ in the definition of $\mathbf{X}$) also influences gradient rank provides directly relevant insights into the effect of that particular hyper-parameter on this phenomenon. It is strictly correct to say that all of these results rely on matrix multiplication; however, AD provides us with the framework to study which matrices are multiplied in which order when computing the gradients via backpropagation.
>
> We have made an argument justifying our choice to start with linear networks in the general response. We believe that it is important in theoretical work to begin from first principles, and to introduce complexity (such as nonlinear activations) only once simplified cases are thoroughly understood. The complexity of an argument alone does predicate its value, and we have argued that even beginning with simplified models and using elementary linear algebraic identities, the insights provided are novel enough to warrant inclusion in this work.

---

> ### Author Response · Authors · 2024-01-07
> **Connection to Neural Collapse**
>
> As we have stated in our general response, the connection between our work and neural collapse comes from our work providing clear boundaries on the dimension of the subspaces which gradients can inhabit during training. Since Neural Collapse describes the reduction of the feature space to a low-dimensional simplex, it makes sense that the geometric properties of the gradient (such as rank, i.e. the size of the subspace the gradients inhabit) would play a role in that behavior.
> Neural Collapse is by definition a dynamical phenomenon, and we make a clear point to steer from dynamics in this work. Instead, our work sets the boundaries within which dynamics can occur, and sets the stage for further work particularly investigating how the rank changes during training. We recognize that it is important to make this distinction clear to our readers, and for this we have expanded our introduction and discussion. Thank you for bringing up this concern!

---

> ### Author Response · Authors · 2024-01-07
> **Significance of Gradient Rank**
>
> In our general response, we have made an argument for why the study of gradient rank matters. Our case relies on two arguments: 1) the determination of gradient rank sets the stage for further study of rank dynamics, 2) direct pragmatic applications which utilize low-rank approximations of the gradient. We are working to expand our introduction, discussion, and related work sections to make the significance more apparent to readers. We thank you for bringing up this concern.

---

### Review · Reviewer_37gU · 2023-12-26

**Summary Of Contributions:**

This paper provides bounds on the rank of the gradients at each iteration of training with mini-batch SGD with various architectures.

**Audience:**

No

**Claims And Evidence:**

No

**Requested Changes:**

Major questions (critical):
-----------

* How does Equation (5) follow from Equation (4)? There is an extra factor of epsilon in Equation (5) that does not seem to make sense there?

* In Figure 4, did you only run the experiment one time? It seems that since the bottleneck is of size 2, the "rank" may vary greatly from run to run.

Minor questions (not critical):
-----------

* On page 4, $\\mathcal{D}\_i$ is introduced as "the error on the output". I'm not sure what this terminology means, apart from the parallel with the definition of $\\delta_i$ for linear networks. I suppose it becomes clear later when you use it to define $\\nabla_{U_i}$ and $\\nabla_{V_i}$.

* What does "correspondences between input" mean on page 5? Do you mean between different inputs $X_t$ and $X_{t+1}$ at different times?

* The bounds on the rank are never explicitly stated in Section 3.2.1, which makes it confusing to later read sentences like "The first thing
to notice is that even for as small as T = 2, we reach the potential for full-rank gradients quite quickly" and "Thus, since the core idea is the same as for RNNs, our theoretical bounds on gradient rank in convolutional layers are included in our supplementary material." Is this omission of the rank bounds for RNNs in Section 3.2.1 accidental?

* On page 5, what is machine-epsilon, numerically? This can vary from architecture to architecture and whether you are using floats or doubles, so it would be good to include for clarity.

* LeakyReLU is never defined in its dependence on $$\alpha$$ until Section 6.1. This makes it difficult to read.

Typos:
--------
* $\Delta_{t,i+1}$ on page 4 in definition of $\mathcal{D}_i$ should be bolded

* "how parameter-tying can improve with parameter-tying" on page 5

* "$r_i(D_{\alpha})$ are the 2-norm also in decreasing order"; the words "of the rows" are omitted

**Strengths And Weaknesses:**

The presentation and flow of the paper is clear. The question studied -- how the rank of the gradients during training depends on architecture and on properties of the data, is interesting.

I think there may be a fairly major mistake in the math -- how does Equation (5) follow from Equation (4)? Specifically, why is there a factor of epsilon in Equation (5)?
I initially thought that this was a typo, but then the "Hypothesis 3" experiments seem to indicate that this epsilon factor is intentional.
Also the Equation (10) in the appendix indicates it as well.

The general example I have in mind that seems to contradict this is:
$$\\mathbf{A} = \\begin{bmatrix} 1 & -1 \\\\ -1 & 1 \\end{bmatrix}$$,
which has rank 1, but if we apply ReLU entrywise we obtain
$$\\mathbf{B} = \\mathrm{ReLU}(\\mathbf{A}) = \\begin{bmatrix} 1 & 0 \\\\ 0 & 1 \\end{bmatrix}$$,
which has rank 2.

---

> ### Author Response · Authors · 2024-01-07
> **Mathematical Correctness**
>
> Thank you for your response! We appreciate that you are being careful with checking our derivation; however, as we will point out now, there is no mathematical mistake in the transition from equation (4) to equation (5).  We have provided a clarification here, demonstrated the correctness using your provided example, and have added additional details to the main text to maximize the clarity.
>
> Let $\mathbf{Z}$ be the internal activations of some layer in a feed forward neural network. Let $\phi(\mathbf{Z})$ be the external activations at that layer. If $\phi_\alpha$ is a leaky-ReLU activation with slope of $\alpha$ in the negative domain, we can equivalently use the Hadamard (elementwise) product to say that $\\phi(\mathbf{Z}) = \mathbf{D}\_{\alpha} \\odot \\mathbf{Z}$, where $\mathbf{D}\_\alpha$ is a matrix containing 1s or $\alpha$ according to the leaky-ReLU activation.
>
> Let $\{\sigma_i\}$ be the set of singular values of the post-activation matrix $\phi(\mathbf{Z}) = \mathbf{D}\_\alpha \odot \mathbf{Z}$. Let $\{\rho_i\}$ be the set of singular values of the internal activations $\mathbf{Z}$. Additionally, let $c_i$ and $r_i$ be the 2-norm of the columns and rows of $\mathbf{D}_\alpha$ respectively, sorted such that $c_1 > c_2 > ... > c_k$ etc.
>
> From Zhan et al. 1997, we have the identity defining a relationship between the singular values of a Hadamard product as
> \begin{equation}
>     \sum_{i=1}^k \sigma_i < \sum_{i=1}^k \min(c_i, r_i) \rho_i
> \end{equation}
>
> This definition, provided as a relationship between sums, does not provide a one-to-one correspondence between the singular values pre and post activation except for when $k=1$. In other words, we only have a direct bound on the largest singular value of $\mathbf{D}_\alpha \odot \mathbf{Z}$ i.e. $\sigma_1$, and the largest singular value of $\mathbf{Z}$ i.e. $\rho_1$:
>
> \begin{equation}
>     \sigma_1 < \min(c_1, r_1) \rho_1
> \end{equation}
>
> An empirical definition of the rank of a matrix can be given in terms of the relationship between its largest singular values, smaller
> singular values, and the machine epsilon which is used to estimate the
> singular values of that matrix. We note that machine epsilon defines
> the precision of a given computing machine for estimating the singular
> values, and is defined such that $0 < \epsilon \le 1$. The definition
> used in Pytorch and in Numerical Linear Algebra at large says that the
> rank of a matrix is the largest index $k$ of the singular values such that
>
> \begin{equation}
>     \sigma_k < \epsilon \sigma_1
> \end{equation}
>
>
> Therefore, we can combine equations (2) and (3) to provide a bound on the largest $k$ $\sigma_k$ which does not contribute to rank as follows:
>
> \begin{equation}
>     \sigma_k < \epsilon \sigma_1 < \epsilon \min(c_1, r_1) \rho_1
> \end{equation}.
>
> ## Example
>
> Let $\mathbf{A} = \begin{bmatrix} 1 & -1\\\\-1 & 1\end{bmatrix}$ and let $\mathbf{B} = \begin{bmatrix}1 & 0\\\\ 0 & 1\end{bmatrix}$. Following our notation above, we have that $\mathbf{B} = \mathbf{D}\_0 \odot \mathbf{A}$, where $\mathbf{D}\_0= \begin{bmatrix}1 & 0\\\\0 & 1\end{bmatrix}$ is the values of the ReLU activation function given the input $\mathbf{A}$. It is clear from the definition of $\mathbf{D}\_0$ in this example, we have $\min(c_1,r_1) = 1$, so the norms of the columns and rows are not needed going forward.
>
> The singular values of $\mathbf{A}$ are $\{\rho\_1, \rho\_2\} = \{2, 0\}$, and the singular values of $\mathbf{B}$ are $\{\sigma\_1, \sigma\_2\} = \{1,1\}$. Therefore, in order to determine the rank of $\mathbf{B}$ using our bound, we can simply check each $\sigma\_k$ in sequence, comparing with the value of $\epsilon \min(c_1, r_1) \rho_1 = \epsilon \rho\_1$
>
> First, we must determine a reasonable value for machine epsilon. For 32-bit floating point numbers in PyTorch, this is defined to be around $1\times 10^{-7}$.
>
> We begin with $\sigma_1 = 1$. We can see that $\sigma_1 = 1 \nless 2\times 10^{-7} = \rho_1 \epsilon$, therefore we say that $\sigma_1$ contributes to the rank of $\mathbf{B}$. Since $\sigma_2 = 1$ as well, it will therefore also contribute to the rank of $\mathbf{B}$. Therefore, the rank of $\mathbf{B}$ is correctly estimated to be 2 according to our bound.

---

> ### Author Response · Authors · 2024-01-07
> **Figure 4 and Repeated Experiments**
>
> As we mention in the last paragraph of section 4, for all experiments we performed 5-fold cross-validation with 20 randomly initialized models, for a total of 100 experiments for each result. Standard deviation bars are in fact actually included in this figure; however, the magnitude is so small that they are difficult to see on individual bars. We have modified the figure so that the deviation in each bar is more apparent.

---

> ### Author Response · Authors · 2024-01-07
> **Minocr Comments**
>
> * The notation $\mathcal{D}_i$ denotes the generalized tensor form of the adjoint variable computed during reverse-mode auto-differentiation, which is nothing more than the partial derivative of the loss function at that layer (accumulated from backprop) w.r.t.\ the internal output activations at that layer. In this case, $\mathcal{D}_i$ is a 3-way tensor, with an additional dimension to account for the accumulation over time. We will add some language upfront in the revision to make this clearer for readers.
>  * Yes, at this point we are referring to correspondences between inputs over time which may affect the final rank of the gradient after accumulating over the temporal dimension. We have changed the language to ``correspondences across time'' in an effort to make this more clear.
>  * Although the omission was not accidental, we recognize that our omission of the linear algebraic identity $\mathsf{rank}(A+B)\le \mathsf{rank}(A) + \mathsf{rank}(B)$ should be included here to make these statements more obvious.
>  * Indeed, machine epsilon does vary from architecture to architecture; however, in numerical linear algebra it is always defined to be a small positive real number, i.e. $\epsilon \in \mathbb{R}, 0 < \epsilon \ll 1$. For example, the machine epsilon value for 32-bit floating point numbers in Pytorch is roughly $1\times 10^{-7}$. We agree that including a clear introduction of machine epsilon helps with the clarity of the work, and we have added this in the revision.
>  * We have moved the introduction of the definition of Leaky-ReLU activations to section 3.3. Thank you for this feedback!

---

> > ### Comment · Reviewer_37gU · 2024-01-10
> > **Still not convinced**
> >
> > I am struggling to make sense of your response, and I think what's going on is that you are actually claiming something much weaker than I think you are trying to claim.
> >
> >
> > Your claim in Equation (9) of the revised text is that for the $i$th singular value we have $$\sigma_i(D_{\alpha} \odot Z) \leq \epsilon \min(c_1(D_{\alpha}), r_1(D_{\alpha}))\sigma_1(Z).$$
> >
> > Clearly this is not the case for all singular values, so I imagine that you are claiming this only for $i$ which are bigger than the numerical rank? If so, this should be specified very clearly in the paper.
> >
> > Once we clear this up, I will give more detailed feedback on the remainder of paper, including experiments.
> > See also my other comments below.
> >
> > ### Other comments
> > * By the way, in your response you write:
> >     > The definition used in Pytorch and in Numerical Linear Algebra at large says that the rank of a matrix is the largest index $k$ of the singular values such that
> > $\sigma_k < \epsilon \sigma_1$
> >
> >     This is incorrect. The correct definition is instead:
> >     > The definition used in Pytorch and in Numerical Linear Algebra at large says that the rank of a matrix is the largest index $k$ of the singular values such that $\sigma_k \geq \epsilon \sigma_1$
> >
> > * Equation (8) of the revised text is missing a parenthesis and seems incomplete -- there also seems to be a missing factor of $\epsilon$ if you really want to get to Equation (9) somehow? But I am not following the argument here.
> >
> > * One modification that would significantly improve clarity is if instead of "rank", the authors wrote "effective rank" or "numerical rank" whenever appropriate.
> >
> > * The variable $k$ could be defined more clearly in the paper. As written, in the text of the paper it is not clear that $k$ is the numerical rank. It is also not clear in Equation (8) of the current manuscript. From the way the text is written it is also not clear that in Equation (9) you take $1 \leq i \leq \mbox{effective rank}$, or what $i$ is?

---

> ### Author Response · Authors · 2024-01-10
> **Further Clarification**
>
> Thank you for your continued discussion! Please allow us to provide further clarification.
>
> As we say before introducing equation 4 in the text:
> "we say that the kth singular value $\sigma_k$ does not contribute to our estimation of rank if $\sigma_k < \epsilon \sigma_1$".
> It is this definition which we are interested in for numerical estimation of rank.
>
> The thrust of our argument in this section amounts to using the bound provided in Zhan et al. to provide a bound on that maximal singular value $\sigma_1$. Using the insight that we can write Leaky-ReLU activations as an elementwise product, we can use the identity from Zhan et al. to give a new bound which depends on the singular values of the internal activations $\mathbf{Z}$ and the coefficients for the Leaky-ReLU $\mathbf{D}_\alpha$.
>
> Of course, then as you say, not all $i$ singular values in equation 9 will fall below this bound. In fact, we need to count the index $i$ which does fall below that bound in order to say that the $i$th singular value does not contribute to estimation of numerical rank.
>
> In our response and in our revised text we are careful to point out that the bound from Zhan et al. can only provide a one-to-one association between the maximal singular values pre and post activation. The claim in equation 9 then is just a restatement of the bound from equation 4, but the right hand side is given in terms of the maximal pre-activation singular value, rather than the post-activation singular value.
>
> In order to find further correspondence between associated singular values (in other words to describe how a general $i$th singular value pre-activation will affect a general $i$th singular value post-activation), we cannot use the identity from Zhan et al, and in fact a general association beyond the maximal singular values does not exist in the literature.
>
> In the text, we have endeavored to be very explicit that we are working with a numerical bound used to estimate rank. The title of the section explicitly states that we are working with "numerical effects" and throughout that section we are careful to say we are working with an "estimation of rank". As you suggest, we think it will help further clarity to explicitly refer to "estimation of numerical rank" in this section to avoid further confusion. Thank you for this helpful suggestion!
>
> W.r.t to the difference in the definitions you have pointed out, we have been providing a definition in line with our description in the text prior to equation 4 that "we say that the kth singular value $\sigma_k$ does not contribute to our estimation of rank if $\sigma_k < \epsilon \sigma_1$". Although it is equivalent to count either the number of singular values that fall strictly below that bound (i.e. we would say the rank is $k-1$ given our definition), or the maximum index $k$ which falls above or onto it, we appreciate you pointing out this ambiguity in our response.
>
> Thank you for pointing our the missing parentheses in equation 8. This was a typo, as was the omission of $\epsilon.$
>
> We have provided a further revision which fixes these typos and refers explicitly to "estimation of numerical rank" as you have suggested.

---

> ### Comment · Reviewer_37gU · 2024-01-27
> **Requested changes**
>
> Sorry for the delay in my reply. I am not comfortable certifying that this paper meets the "Claims and Evidence" standard. Several changes would have to be made before to make it clearer and ensure correctness.
>
> 1. **Notational change for rank**. The current notation uses "rank(A)" to refer to both the standard notion of rank of a matrix (e.g., in Sections 3.1 and 3.2) and also to the "numerical rank" of a matrix (e.g., Section 3.3.1). This was a big source of confusion for me.
>
>     Please use notation that explicitly differentiates between "numerical rank" and the standard notion of rank. For example, you could use the notation "$\\mathrm{rank}\_{\\epsilon}$" or "$\\mathrm{nrank}\_{\\epsilon}$" for numerical rank, which also has the benefit of making clear the dependence on $\\epsilon$, and also disambiguates the different sections.
>
>     Please also avoid using the shorthand "rank" when you mean to refer to "numerical rank". You could call it "n-rank" or "$\\epsilon$-rank" if you want a shorthand, but it's a source of confusion otherwise.
>
> 2. **Clarify and correct Section 3.3.1**. I understand now from your response that you mean in Equation (9) that you only mean for this to apply to $i$ which are greater than the numerical rank.
>
>     However, this is not actually stated anywhere in this section. In fact, several equations in this section are currently quite unclear, and clarity/correctness can be vastly improved with small changes.
>
>     I will give some examples of what I mean:
>     * "If we can find such a k, we would therefore conclude that the rank of the matrix is k − 1". This is incorrect. Instead, what you can conclude is that the "**numerical** rank of the matrix is **at most** k-1".
>
>         I would suggest instead to use this as place to define the notation for numerical rank with a statement like: "We denote the numerical rank of a matrix $M$ by $\\mathrm{nrank}\_{\\epsilon}(M) = \max \\{k : \\sigma\_k(M) \geq \\epsilon \\sigma\_1(M)\\},$ where $\\sigma\_1(M) \\geq \\sigma\_2(M) \\geq \\dots $ are the singular values of $M$.
>
>     * The derivation from equation (6) to equation (7) is not actually needed here. In Zhan [1997], Lemma 2(c) is the statement that $\\sigma\_i(A \\odot B) \leq \\min(c_i(A), r_i(A)) \\sigma_1(B),$ for any i. Specializing to the case of $i = 1$ yields $\\sigma\_1(A \\odot B) \leq \\min(c_1(A), r_1(A)) \\sigma_1(B),$ which is equation (7).
>
>       So in no place do you need the sum in Equation (6) which adds confusion because it is not clearly stated what $K$ is -- in one place it is taken to be $\\min(N,m)$, and in another it is taken to be $\\min(N,h)$, and in another it is taken to be $K = 1$.
>
>     * In Equation (9), please be more explicit about what $i$ is. It is never explicitly stated. A statement like, "for $i > \\mathrm{nrank}\_{\\epsilon}(D\_{\\alpha} \\odot Z)$ would make this equation understandable.
>
>     * In the explicit expressions for $c\_i(D\_{\\alpha})$ and $r\_i(D_{\\alpha})$, please be explicit about dependence of $N_+$ and $N_-$ and $M_+$ and $M_-$ on $i$. E.g., you could use the notation $N_+(i)$ or $N_{+,i}$. Otherwise, it seems like you are proving a bound where the right-hand-side is independent of $i$.
>
>     * The definition "where $N_-$ is the number of rows in column $i$ ... " is incorrect. It is not the $i$th column -- the values of $N\_{+,i}$ have to be sorted so that $N\_{+,1} \\geq N\_{+,2} \\geq \\dots $ etc...
>
>     * There are parts of the Appendix C, whose objective is to extend the results in Section 3.3.1, that do not make sense. E.g., $k$ is not defined in equations (13) and (14), and it is unclear how (14) is derived. And also this appendix does not prove what is claimed in the main text, which claims that the analyses of Section 3.3.1 "can be applied to any piecewise linear function".  Instead, this appendix is about the singular values for the matrix of partial derivatives, instead of the activation matrix.
>
>
> Please also fix typos (these occur throughout the paper, but I checked Section 3.3.1 most carefully):
> * On page 6, "<<" should be written in LaTeX  as "\\ll" instead.
> * On page 6, "weight matrix $W \\in $" .... is misrendered
> * On page 6, before Equation (5): Leaky-ReLUiactivations
> * On page 6, "producut"
> * On page 6, Equation (6) has $\\sigma_i(Z_i)$, but instead $\\sigma_i(Z)$ is correct.
> * On page 7, "the affect" should be "the effect"
>
> ---------
>
> As far as whether there is an audience for this paper, the general research direction of low-rankness in gradients and weights of a neural network is very interesting. However, the theoretical results in this paper in Section 3 do not seem to add new insights to the field. The arguments in Section 3.1 and 3.2 fall into a similar style of known folklore results since they follow from
> $\\mathrm{rank}(AB) \\leq \\min(\\mathrm{rank}(A),\\mathrm{rank}(B))$. The bound in Section 3.3 is direct from past work and it is unclear what new elements are appearing here.

---

### Author Response · Authors · 2024-01-07
**Overall Response - Part 1**

We thank the reviewers for their insightful feedback. We value your input and have begun revising our piece to address your critiques. In an effort to better elucidate our contributions to the study of gradient rank in deep neural network training dynamics, we've refined technical details, and are working further to enhance the introduction, related work, and discussion sections. We are committed to clarifying the novelty and community value of our work for TMLR's readers. We appreciate your highlighting areas for expansion and clarification.

The major concern put forward by two of the reviewers deals with the novelty of the insights we provide in this paper, particularly with respect to the bounds on gradients in fully linear networks. Because this concern was brought up by two reviewers, and because we believe it is critical to assuage this concern immediately, we would like to include a response to this claim in our general response.

Our choice to begin theoretical analysis with linear networks is based off of previous work in the literature, particularly multiple works from Saxe et al. in which the authors derive exact equations for training dynamics. The reason provided in these works is that the introduction of nonlinear activations significantly complicates the derivation of any solvable dynamics. In addition, the results provided by the authors of these work demonstrate that the solutions to the derived systems of equations provide reasonable approximations to observed empirical behavior in networks with ReLU nonlinearities. Our choice to begin with linear networks follows a similar argument - we believe that it is important to start from first principles, from simple systems in which standard mathematical tools provide clear results which are simple to understand. We believe that it is important to establish clear groundwork before moving into the more complicated results which arise from the introduction of additional details like nonlinear activations, batch normalization, etc.  In short, the choice to begin with linear networks is a didactic one, and the simplicity of our argument in the initial theoretical sections of our work is not lost on us; however, we believe that the straightforward nature of an argument should not detract from it (the alternative is to favor complex arguments simply by nature of their complexity). Furthermore, as we will argue next, we believe that the claim of triviality omits some of the details of our argument which emerge from out unique vantage point in auto-differentiation.

The bound which we derived for the gradient is given in equation (5) as

$$\\min \\{ \\mathsf{rank}(\\mathbf{X}),\\mathsf{rank}(\\mathbf{W}_1),...,\\mathsf{rank}(\\mathbf{W}_i),...\\mathsf{rank}(\\mathbf{W}_L),rank(\\partial\\mathcal{L}/\\partial\\mathbf{Z}_L)\\}$$

We can immediately see that the ranks of the gradients are **in fact not just limited by the shape of the weight matrices**, but also by the rank (not just the shape) of the input and the first derivative of the loss function. Importantly, we obtain this insight by observing how the adjoint matrices are computed during the backward pass in reverse-mode back-propagation, not via a simple forward multiplication. In fact, the backward pass in reverse-mode AD will cause rank bottlenecks at deeper layers to move upward through the network, an insight which we believe is important as it is extremely common to utilize classifier modules with very few neurons at the output layer. The insight that the number of classes for example limits the rank of the gradients **in the entire network** naturally emerges from our result, and we believe it illustrates the importance of making this fact recognized to the community even in the simple case of linear networks. Additionally, we have pointed out in our discussion on AD that all gradient ranks will also be limited **by the size of the batch** used for each update - this insight which also naturally emerges from our discussion in sections 3.1 and 3.2 establishes an important question: how might a small batch affect the dynamics of learning due to constraining the gradient to a small-dimensional subspace? Although addressing such a question is outside the scope of this particular work, it demonstrates that our insights resonate beyond the simple recognition that matrix multiplication will result in limited rank. Although we do use this elementary linear algebraic result to derive the bound, the insights provided from our perspective in Auto-Differentiation elevate the ramifications beyond statement of trivial facts.

---

### Author Response · Authors · 2024-01-07
**Overall Response - Part 2**

At last, we would like to address the question of why gradient rank is interesting to the TMLR community in the first place. First, we would like to cement the argument connecting this work to the phenomenon of Neural Collapse.  In Neural Collapse, a collapse of the feature spaces to a low-dimensional simplex (not a single point) is observed during overparametrization in classification tasks. As the weights in a neural network are updated by the addition of computed gradients, it seems obvious that the dimensionality of gradients should play some role in guiding that movement toward a particular low-dimensional geometric structure. Particularly, if the gradients remain full rank throughout the network, it would seem strange to observe a collapse into a low-dimensional feature space. At its core, Neural Collapse is a dynamical behavior, and although we make a clear point to avoid any direct discussion of dynamics in this work, what our work does is establish how the architecture of a network prior to training can determine the dimensionality of gradients. Establishing this framework is an important prior step to analyzing how the dynamics of rank change during training, and we believe the connections to Neural Collapse will be fully realized once this step is taken.

Beyond Neural Collapse, our study of gradient rank has relevant to the results in Saxe et al., in which the derived dynamics for linear networks demonstrate that ``dominant modes'' (i.e. the largest singular values of input-output covariance) are learned in sequence during training. Although our work is not itself dynamical, as with Neural Collapse, our results set the stage for these kinds of results entirely based off the architecture itself, particularly our bounds exactly determine how many singular values will even be available throughout the linear networks discussed in Saxe's work. In sum, although our paper is not itself a paper on dynamics, it establishes important boundaries for the dynamics, especially in the sense that these dynamics have some kind of dimensional relevance (collapsing to a low-dimensional structure, learning dominant singular values in sequence, etc).

Finally, we would like to point out that there are important applications of the gradient rank beyond learning dynamics in deep neural networks. One entirely pragmatic application comes from work on federated learning, particularly in methods such as PowerSGD \cite{vogels2019powersgd} which use low-rank approximations of the gradients in order to lower communication costs in federated optimization. In such methods, the rank utilized for these approximations is a hyper-parameter, and although there are clear empirical trade-offs in using lower-rank approximations, there hasn't been any principled theoretical discussion investigating how much exactly is lost by choosing a lower rank. In this work we can provide exact guidance for linear networks in which rank can be exactly determined. We push beyond this by beginning the more subtle work of introducing nonlinear activations, demonstrating that for Leaky-ReLU style activations the choice of machine epsilon and the level of nonlinearity in the negative domain may increase rank, thus resulting in the potential for more information loss from low-rank approximations.

We believe that these general responses should help to cement the importance of our work, and address concerns that the insights we provide are somehow too trivial or not relevant to the TMLR community. Since these concerns we brought forward by the reviewers, we have seen there is a clear need to make these arguments clearer to readers in the main paper.  We have begun this work in our revisions, expanding our introductions, related work, and discussion sections to make sure the relevance and importance is not lost on TMLR's readership.

---

### Decision · Action_Editor_xfBr · 2024-02-01

**Recommendation:** Reject

**Comment:**

This paper presents a theoretical and empirical investigation into the impact of network architecture and activation function choices on the rank of gradients in Deep Neural Networks (DNNs). Reviewers acknowledge that the research direction is interesting, and appreciate the extensive experiment works. Nonetheless, reviewers have also raised several significant issues, including unclear notations and derivations, blurred contributions, and results that appear somewhat trivial. During the rebuttal phase, the authors engaged in thorough discussions with reviewers and made commendable efforts to clarify these issues. Despite these efforts, concerns remain about the clarity of the notations and derivations (as noted by Reviewer 37gU), as well as the trivial nature of the results, which may not provide significant value to the audience (as noted by Reviewer Rv5s). Due to these unresolved issues, it is recommended that the authors undertake a major revision to address these critical concerns before resubmission.

**Audience:**

Reviewers feel that the research topic explored in this paper is intriguing. However, there are reservations about the paper's overall contribution. Reviewer Rv5s perceives the contributions as somewhat trivial, and reviewer 37gU expressed concerns about the theoretical results in their official response. These collective viewpoints cast doubt on the work's interest to the ML community.

**Claims And Evidence:**

Reviewers have expressed concerns about the clarity of its claims and supporting evidence. Reviewer 37gU specifically points out issues with unclear notations and derivations, which may lead to potential misunderstandings. Additionally, Reviewer Rv5s has raised similar concerns. These points suggest a need for more detailed clarification and robust evidence to strengthen the work's credibility.

**Resubmission Of Major Revision:**

The authors may consider submitting a major revision at a later time.